# AGS3 antagonizes LGN to balance oriented cell divisions and cell fate choices in mammalian epidermis

Carlos P Descovich[1,2], Kendall J Lough[1], Akankshya Jena[1], Jessica J Wu[1], Jina Yom[1], Danielle C Spitzer[1†], Manuela Uppalapati[1], Katarzyna M Kedziora[3], Scott E Williams[1,4]*

[1]Department of Pathology and Laboratory Medicine and Biology, University of North Carolina at Chapel Hill, Chapel Hill, United States; [2]Curriculum in Cell Biology & Physiology, Lineberger Comprehensive Cancer Center, University of North Carolina at Chapel Hill, Chapel Hill, United States; [3]Bioinformatics and Analytics Research Collaborative, University of North Carolina at Chapel Hill, Chapel Hill, United States; [4]Lineberger Comprehensive Cancer Center, The University of North Carolina at Chapel Hill, Chapel Hill, United States

**\*For correspondence:**
scott_williams@med.unc.edu

**Present address:** [†]Department of Molecular and Cellular Biology, University of California, Berkeley, Berkeley, United States

**Competing interest:** The authors declare that no competing interests exist.

**Abstract** Oriented cell divisions balance self-renewal and differentiation in stratified epithelia such as the skin epidermis. During peak epidermal stratification, the distribution of division angles among basal keratinocyte progenitors is bimodal, with planar and perpendicular divisions driving symmetric and asymmetric daughter cell fates, respectively. An apically restricted, evolutionarily conserved spindle orientation complex that includes the scaffolding protein LGN/Pins/Gpsm2 plays a central role in promoting perpendicular divisions and stratification, but why only a subset of cell polarize LGN is not known. Here, we demonstrate that the LGN paralog, AGS3/Gpsm1, is a novel negative regulator of LGN and inhibits perpendicular divisions. Static and ex vivo live imaging reveal that AGS3 overexpression displaces LGN from the apical cortex and increases planar orientations, while AGS3 loss prolongs cortical LGN localization and leads to a perpendicular orientation bias. Genetic epistasis experiments in double mutants confirm that AGS3 operates through LGN. Finally, clonal lineage tracing shows that LGN and AGS3 promote asymmetric and symmetric fates, respectively, while also influencing differentiation through delamination. Collectively, these studies shed new light on how spindle orientation influences epidermal stratification.

## Editor's evaluation

Descovich et al. examine the important decision between proliferative (planar) and differentiation (perpendicular) divisions in the basal layers of the skin and find a key promoter of perpendicular divisions is inhibited by its paralog to specify planar divisions. The authors use sophisticated mouse genetics and imaging and discover that LGN and its paralog AGS3 function antagonistically in determining perpendicular vs. planar divisions. The claim that AGS3 functions as a natural dominant negative version of LGN to control division orientation is well supported.

## Introduction

Asymmetric cell divisions (ACDs), whereby progenitor cells divide to give rise to daughter cells that adopt different fates, are an important mechanism to promote cellular diversity. Many ACDs rely on proper orientation of the mitotic spindle, which can be influenced by intrinsic and extrinsic cues

as well as mechanical forces (*Finegan and Bergstralh, 2019*; *Lechler and Mapelli, 2021*; *Sunchu and Cabernard, 2020*; *van Leen et al., 2020*; *Venkei and Yamashita, 2018*; *Williams and Fuchs, 2013*). Spindle orientation is regulated intrinsically by an evolutionarily conserved ternary complex, which includes the core scaffolding protein LGN (Gpsm2), microtubule-binding protein NuMA/Mud, and small G proteins of the Gαi/o family (*Colombo et al., 2003*; *Du and Macara, 2004*; *Du et al., 2001*; *Schaefer et al., 2000*). In *Drosophila*, the LGN ortholog Pins plays a key role in neuroblast stem cells, where it orients the mitotic spindle to promote the unequal inheritance of fate determinants that results in asymmetric daughter cell fates (*Bellaïche et al., 2001*; *Schaefer et al., 2000*). The Par3/aPKC/Par6 polarity complex and Insc—among other proteins—facilitate membrane association of the LGN–NuMA–Gαi ternary complex, while downstream, the motor protein dynein mediates pulling forces on astral microtubules to reorient the spindle (reviewed in *Bergstralh et al., 2013*; *di Pietro et al., 2016*; *Morin and Bellaïche, 2011*; *Tadenev and Tarchini, 2017*).

In developing stratified epithelia, we and others have shown that, in a subset of basal progenitors, LGN localizes asymmetrically to the apical cortex and promotes perpendicular divisions (*Lechler and Fuchs, 2005*; *Luxenburg et al., 2011*; *Williams et al., 2011*; *Williams et al., 2014*). Epidermal loss of LGN, or deletion of NuMA's microtubule-binding domain, leads to reduced perpendicular divisions, decreased differentiation, and impaired barrier function, resulting in neonatal lethality (*Seldin et al., 2016*; *Williams et al., 2011*). These studies highlight the critical importance of perpendicular divisions in establishing proper epidermal architecture. However, only half of mitoses result in perpendicular divisions in wild-type (WT) embryos, with the other half occurring at an orthogonal, planar orientation. LGN is absent from the cortex in planar divisions, and planar divisions occur independently of known spindle orienting proteins, including LGN, NuMA, Insc, Par3, and Gai3 (*Williams et al., 2011*; *Williams et al., 2014*). This raises the related questions of whether planar divisions are simply a default process, and if the ability of only some cells to polarize apical LGN is actively regulated.

LGN/Gpsm2 is a member of the AGS (activator of G-protein signaling) family of proteins, named because they promote G-protein signaling in a receptor-independent manner. In vertebrates, LGN has a closely related paralog, AGS3 (Gpsm1), and the two proteins share high protein homology and a conserved domain structure, consisting of seven to eight N-terminal tetra-tricopeptide repeats (TPRs) and four C-terminal GoLoco motifs—also known as the G-protein regulatory (GPR) region—separated by a flexible linker (*Blumer et al., 2005*; *Schiller and Bergstralh, 2021*; *Wavreil and Yajima, 2020*). Biochemical and structural studies predict that AGS3 retains the ability to interact with many of the same binding partners as LGN, including Insc and NuMA (*Adhikari and Sprang, 2003*; *Culurgioni et al., 2011*; *Izaki et al., 2006*; *Saadaoui et al., 2017*; *Yuzawa et al., 2011*; *Zhu et al., 2011b*).

However, there remains sparse evidence that AGS3/Gpsm1 possesses spindle orienting activity. An early study showed that AGS3 loss in murine ventricular zone neuronal progenitors increased the proportion of planar divisions, a phenotype that could be mimicked by overexpressing Gαi3 or blocking Gβγ signaling (*Sanada and Tsai, 2005*). On the other hand, another study found that LGN, but not AGS3, was expressed in ventricular zone progenitors and suggested that RNAi targeting of AGS3 had no spindle orientation phenotype (*Konno et al., 2008*). Most recently, the Morin lab showed that *Gpsm1*$^{-/-}$ knockout brains have normal spindle orientation (*Saadaoui et al., 2017*). Overall, while AGS3 does not appear to regulate division orientation in the developing brain, LGN has a well-documented role in promoting planar divisions (*Fujita et al., 2020*; *Konno et al., 2008*; *Lacomme et al., 2016*; *Mora-Bermúdez et al., 2014*; *Morin et al., 2007*; *Saadaoui et al., 2017*).

The canonical spindle orientation machinery, including LGN, has long been assumed to operate exclusively during metaphase to orient division prior to chromosome segregation. However, we recently described a novel late-stage spindle orientation process that shares similarity to the post-mitotic reintegration described in *Drosophila* egg chamber follicular epithelium and murine intestinal crypts (*Bergstralh et al., 2015*; *Cammarota et al., 2020*; *McKinley et al., 2018*). We found that a high percentage of epidermal basal cells enters anaphase at oblique angles, which later correct to either planar or perpendicular over the next hour, with the majority of correction occurring in the first 10–15 min after anaphase onset (*Lough et al., 2019*). We have termed this process 'telophase correction' because it initiates during telophase. While this mechanism is dependent on cell–cell adhesions and their actin scaffolds, it remains unclear whether LGN or other spindle effectors participate in telophase correction or are truly exclusive to early spindle positioning.

Here, we further investigate the role of AGS3 and LGN in regulating oriented cell divisions during epidermal stratification. We find that AGS3 favors planar divisions by antagonizing the ability of LGN to promote and maintain perpendicular divisions. Both knockdown and global knockout of AGS3/ Gpsm1 lead to a higher proportion of perpendicular divisions during peak stratification, while overexpression increases planar divisions. AGS3 loss enhances apical LGN localization throughout mitosis, while AGS3 overexpression decreases the efficiency of LGN polarization. Using ex vivo live imaging of epidermal explants, we show that LGN not only regulates initial perpendicular spindle orientations during early stages of mitosis but also plays a maintenance role during telophase, promoting perpendicular reorientation. In support of a model where AGS3 opposes—and acts through—LGN throughout mitosis, we find that AGS3 promotes planar divisions yet has no effect when LGN is absent. Finally, using mitotic clone genetic lineage tracing in AGS3 and LGN-deficient mice, we show that impairing perpendicular and planar divisions, respectively, has direct effects on asymmetric and symmetric cell fates, while also indirectly impacting the other major route of differentiation in the epidermis, delamination. Together, these data suggest that the two vertebrate Pins orthologs LGN and AGS3 play opposing roles in regulating spindle orientation and differentiation in the developing epidermis.

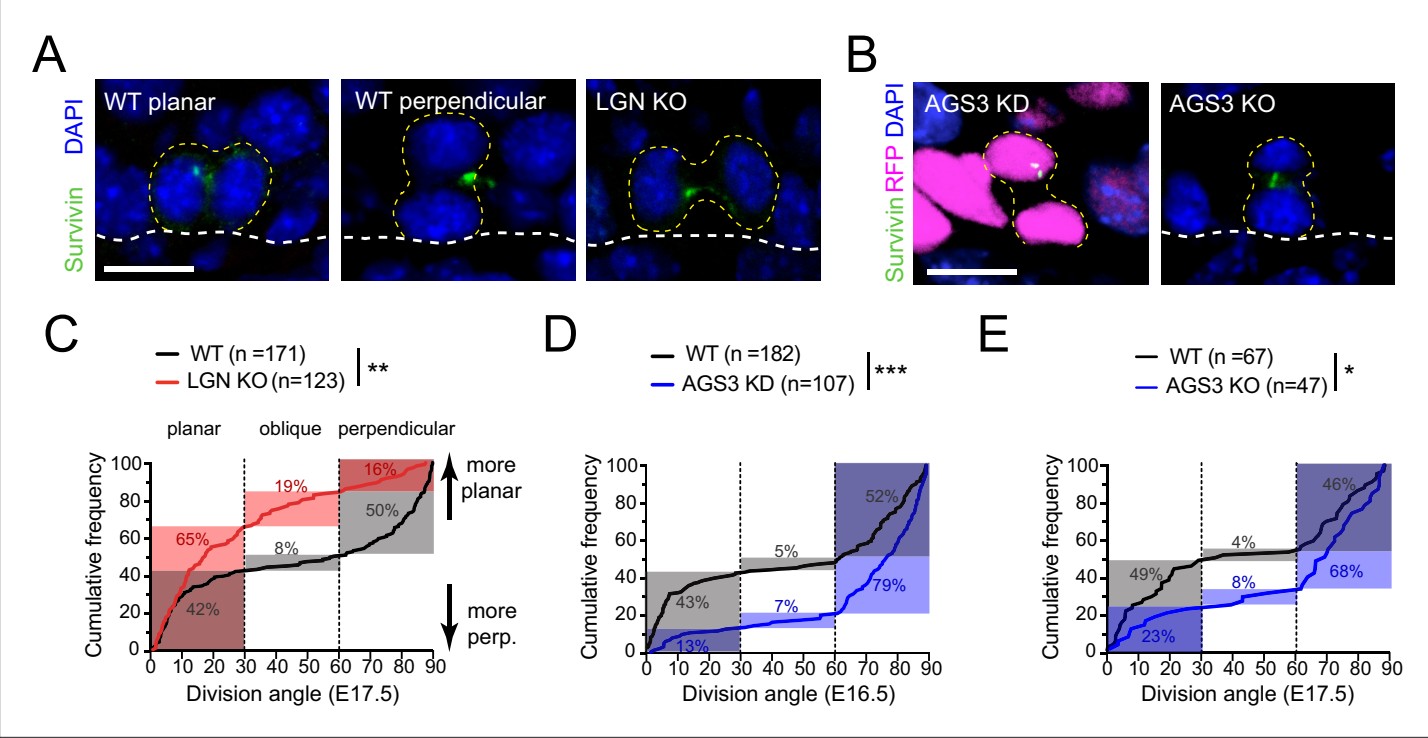

**Figure 1.** LGN/Gpsm2 and AGS3/Gpsm1 have opposing effects on division orientation. (**A**) Images from embryonic day (E) 17.5 sagittal sections stained for Survivin (green), a late-stage mitotic marker that localizes to the spindle midbody during telophase, showing wild-type (WT) planar and perpendicular divisions, and LGN KO (*Gpsm2*$^{-/-}$) planar division. (**B**) Images as in (**A**) from AGS3 KD (*Gpsm1*$^{1147}$ shRNA) and AGS3 KO (*Gpsm1*$^{-/-}$). (**C**) Cumulative frequency distribution of terminal division angles from fixed E17.5 sections of WT littermate controls and LGN KO embryos. Planar (0–30°), oblique (30–60°), and perpendicular (60–90°) bins are shown by dashed lines. Shaded areas indicate the proportion of divisions occurring within each bin for WT (black) and *Gpsm2*$^{-/-}$ KO (red) cells. The upward shift in the LGN KO curve reflects increased planar divisions. (**D**) Cumulative frequency distributions as in (**C**) for E16.5 *Gpsm1*$^{1147}$ knockdown (blue) and uninjected control littermates (black). (**E**) Cumulative frequency distributions for E17.5 AGS3 KO (blue) and WT littermates (black). Scale bars: 10 μm. *p < 0.05, **p < 0.01, and ***p < 0.001 by Kolmogorov–Smirnov test. *n* values (parentheses) indicate cells from >4 embryos per genotype.

# Results

## LGN/Gpsm2 and AGS3/Gpsm1 have opposing functions in oriented cell divisions

Proper orientation of the mitotic spindle relies on the evolutionarily conserved ternary protein complex LGN–NuMA–Gαi (reviewed in *Bergstralh et al., 2017*; *Lechler and Mapelli, 2021*; *Morin and Bellaïche, 2011*). During epidermal stratification, this complex localizes to the apical cortex in 50–60% of mitotic basal keratinocytes where it promotes perpendicular divisions (*Lechler and Fuchs, 2005*; *Williams et al., 2011*; *Williams et al., 2014*). Previously, we have shown that lentiviral knockdown of LGN results in a loss of perpendicular divisions during peak stratification (*Lough et al., 2019*; *Williams et al., 2011*). Here, using the spindle midbody protein Survivin to label late-stage (anaphase–telophase) mitotic cells (*Figure 1A, B*), we characterized terminal division orientation in embryonic day (E) 17.5 back skin epidermis in several loss-of-function models for LGN (*Gpsm2*) and AGS3 (*Gpsm1*).

When plotted as a cumulative frequency histogram, WT basal cell division angles show a characteristic inverted sigmoid pattern. Most divisions fall within either the planar (0–30°) or perpendicular (60–90°) bins with very few oblique (30–60°) divisions, which accounts for the steep slopes at the beginning and end of the cumulative frequency plot, and the relatively flat slope in the middle portion (*Figure 1C*, black line). By comparison, the cumulative frequency plot for *Gpsm2*$^{-/-}$ knockouts (hereafter, LGN KOs) is shifted upward (*Figure 1C*, red line), reflecting an increase in planar and oblique divisions and a sharp reduction in perpendicular divisions, relative to WT controls. The proportions of divisions for each genotype that fall within each orientation bin (planar, oblique, and perpendicular) are highlighted by the gray (WT) and red (LGN KO) shaded regions in *Figure 1C*. In agreement with our previous analyses of *Gpsm2*$^{1617}$ knockdowns (*Williams et al., 2011*), germline genetic deletion of *Gpsm2* in LGN KOs leads to a significant increase in planar (0–30°) orientations, confirming that LGN is important for perpendicular divisions in the developing epidermis.

Despite sharing high homology with its paralog LGN, whether AGS3 possesses spindle orienting activity of its own remains controversial, and two studies in the developing brain have come to different conclusions (*Saadaoui et al., 2017*; *Sanada and Tsai, 2005*). The Morin lab has shown that AGS3 cannot rescue the spindle orientation phenotype caused by LGN loss, while we previously showed that AGS3 loss does not enhance the epidermal thinning phenotype caused by LGN loss (*Saadaoui et al., 2017*; *Williams et al., 2011*). Collectively, these data suggest that AGS3 and LGN are not functionally redundant but leave unresolved the issue of whether AGS3 plays any role in oriented cell divisions.

To test this, we performed ultrasound-guided in utero lentiviral injection (*Beronja et al., 2010*) to transduce embryonic epidermis with a construct that expresses a validated *Gpsm1*$^{1147}$ shRNA (*Williams et al., 2011*). This method allows high-efficiency (>90%) transduction of surface epithelia using vectors that simultaneously express shRNAs via U6 promoter and fluorescent reporters such as histone H2B-mRFP1 or cDNAs via PGK promoter (*Byrd et al., 2016*; *Dor-On et al., 2017*; *Lough et al., 2019*; *Lough et al., 2020*; *Luxenburg et al., 2011*; *Luxenburg et al., 2015*; *Sandoval et al., 2021*; *Williams et al., 2011*). Upon AGS3 knockdown (AGS3 KD), we observed a significant downward shift in the cumulative frequency distribution of oriented cell divisions compared to WT littermates (*Figure 1D*). In AGS3 KD epidermis, perpendicular division frequency increased from 52 to 79%, while planar division frequency decreased from 43 to 13%. Similarly, in germline *Gpsm1*$^{-/-}$ knockouts (hereafter, AGS3 KOs), perpendicular divisions are more numerous (68 vs. 46% in WT littermates) and planar divisions more infrequent (23 vs. 49%, *Figure 1E*). Collectively, these data demonstrate that AGS3 promotes planar divisions during epidermal stratification.

## The LGN/Gpsm2 paralog AGS3/Gpsm1 is mostly cytoplasmic during mitosis

Previous studies across different epithelial tissues have shown that that the subcellular localization of LGN is context dependent (*Ballard et al., 2015*; *Byrd et al., 2016*; *Lacomme et al., 2016*; *Peyre et al., 2011*; *Williams et al., 2011*; *Williams et al., 2014*). While a recent study has shown that AGS3 is cytoplasmic in mitotic neuronal progenitors (*Saadaoui et al., 2017*), we took

a multi-pronged approach to ascertain the subcellular localization of AGS3 in mitotic epidermal basal cells using immunohistochemistry, lentiviral-mediated expression of tagged proteins, and live imaging.

A lack of antibodies specific to AGS3 made it challenging to visualize endogenous protein in fixed tissue. However, we were able to circumvent this issue by comparing the staining patterns observed with a validated affinity-purified guinea pig anti-LGN antibody (*Williams et al., 2011*) and a rabbit pan-LGN/AGS3 antibody (*Williams et al., 2014*) in WT and LGN (*Gpsm2*) knockout mice. While the latter antibody is commercially available and reported to be LGN specific, another group has suggested it can also recognize AGS3 (*Chishiki et al., 2017*), and similar antibodies raised to the C-terminus of LGN also recognize AGS3 (*Konno et al., 2008*). Both antibodies detect an apical crescent in WT mitotic E16.5 basal progenitor cells (*Figure 2A*, left), consistent with the reported localization of LGN (*Lechler and Fuchs, 2005*; *Lough et al., 2019*; *Luxenburg et al., 2011*; *Williams et al., 2011*; *Williams et al., 2014*). However, in mice lacking LGN (*Tarchini et al., 2013*), nearly all specific staining was lost for the guinea pig antibody, while cytoplasmic staining remained in mitotic—but not interphase—cells stained with the rabbit antibody (*Figure 2A*, right). As further confirmation that the rabbit antibody recognizes AGS3 in vivo, we demonstrated that (1) it labels tagged, overexpressed AGS3 and (2) the mitotic cytoplasmic signal we observe in LGN-deficient epidermis disappears when AGS3 is also deleted (*Figure 2—figure supplement 1A, B*). Thus, we observed primarily cytoplasmic localization of endogenous AGS3 during mitosis in the developing epidermis.

As a second approach, we used our in utero lentiviral transduction technique to express epitope-tagged AGS3 in the developing epidermis. First, we overexpressed N-terminal mRFP1-tagged AGS3 (mRFP1-AGS3) in WT embryos and detected its expression in E16.5 dorsal epidermis. This method again revealed cytoplasmic localization in the majority (87%, $n = 54$) of RFP+ mitotic cells, not seen in uninjected littermates (*Figure 2B, C*). In fixed tissue, overexpressed mRFP1-AGS3 could be detected through all phases of mitosis (*Figure 2—figure supplement 1C*).

Because the addition of a tag to the N-terminus—where Insc, NuMA, and other binding partners interact via the TPRs—could interfere with the normal localization or function of AGS3, we also created a lentivirus where AGS3 was tagged with the small V5 epitope at its C-terminus. Since this construct also contains the *Gpsm1*[1147] shRNA, AGS3-V5 should constitute the majority of detectable AGS3 in transduced tissue, thus mitigating potential overexpression artifacts. Immunostaining with a V5 antibody again revealed cytoplasmic localization during mitosis (*Figure 2D*).

In order to track the subcellular localization of AGS3 throughout cell division we utilized ex vivo live imaging of embryonic epidermal explants (*Cetera et al., 2018*; *Lough et al., 2019*). We utilized in utero lentiviral transduction to express an shRNA-resistant C-terminal mKate2-tagged AGS3 (AGS3-mKate2) together with the *Gpsm1*[1147] shRNA to knockdown endogenous AGS3. These experiments were performed on a *Krt14*[Cre]; *Rosa26*[mTmG] reporter background, which ensured that epidermal cells were membrane-GFP (mG)+, while *Krt14*-negative tissues (such as dermis and melanocytes) remained membrane-Tomato (mT)+. To minimize phototoxicity, the epidermis was imaged at low laser power settings on a Dragonfly spinning disc confocal, necessitating implementation of Noise2Void (N2V), a deep-learning (DL) image denoising method which utilizes self-supervised training (*Krull et al., 2020*; *Figure 2E*). Once again, in epidermal basal cells we observed AGS3-mKate2 predominantly in the cytoplasm throughout mitosis (*Figure 2F*).

## AGS3 overexpression increases planar divisions and disrupts apical LGN

Since AGS3 is predominantly cytoplasmic, and loss of AGS3 and LGN has opposing effects on oriented cell divisions, we wondered whether AGS3 might negatively regulate LGN's activity by sequestering it away from the apical cell cortex. As a first test of this hypothesis, we overexpressed AGS3 using our N-terminal tagged mRFP1-AGS3, delivered by in utero lentiviral transduction. In epidermis from transduced embryos, we quantified division orientation using Survivin as a late-stage mitotic marker (*Figure 3A*). Cumulative frequency histograms show that when mRFP1-AGS3 is expressed at E16.5—an age where most divisions are typically perpendicular (e.g., 52% in *Figure 1D*)—the majority of divisions (56%) were instead planar (*Figure 3B*). However, since the effect of AGS3 overexpression with this construct was mild, we considered that the N-terminal RFP tag might interfere with its TPR-binding function.

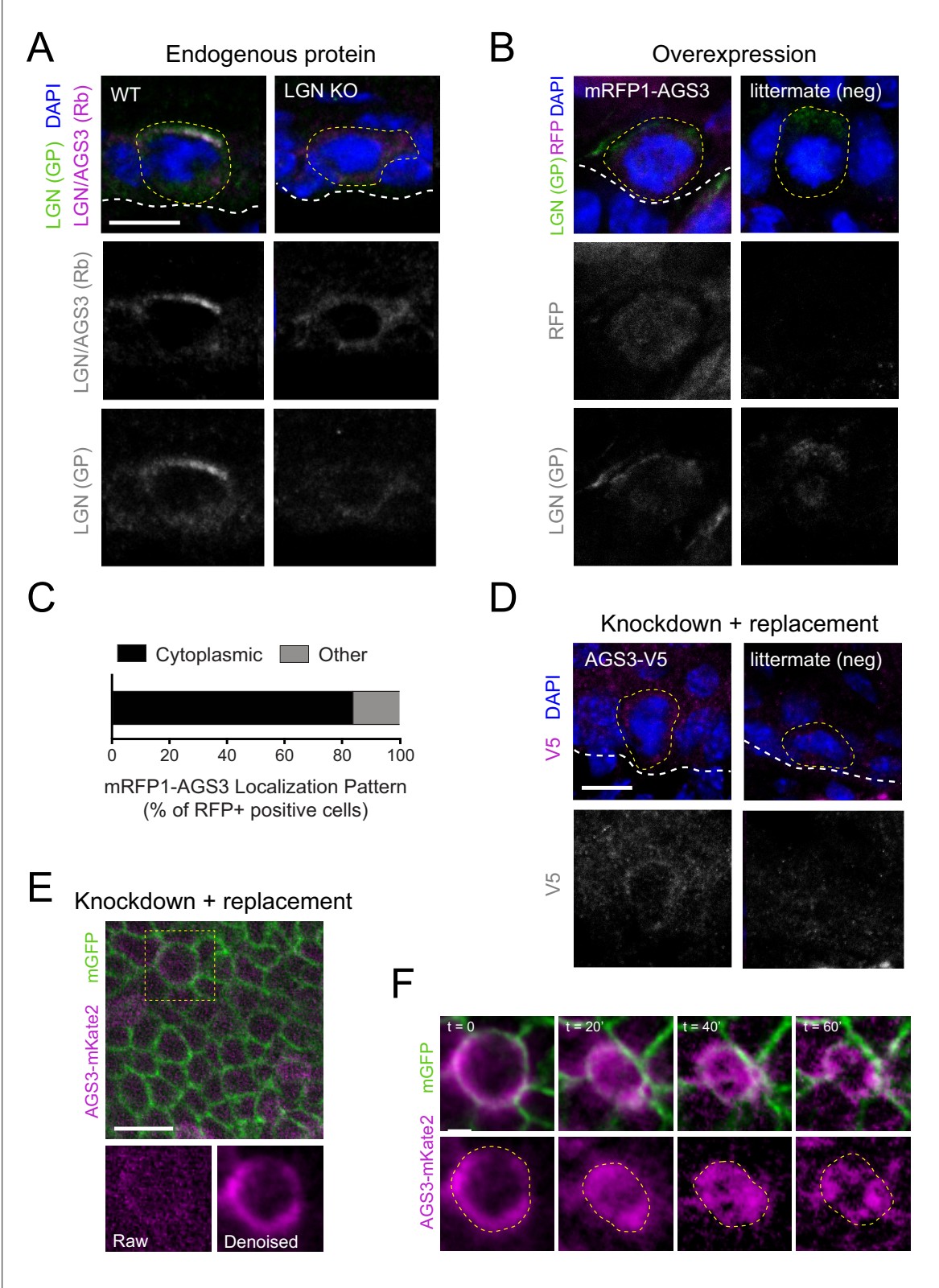

**Figure 2.** AGS3 localizes to the cytoplasm during mitosis. (**A**) Immunofluorescent images from E17.5 sagittal sections of wild-type littermate controls (left) and LGN KO (*Gpsm2*$^{-/-}$) embryos (right) showing mitotic basal cells stained for guinea pig (GP) anti-LGN (green) and rabbit (Rb) anti-LGN/ AGS3 (magenta). Merged images shown at top and single channels below. Cortical signal is lost in LGN KOs while LGN/AGS3 shows cytoplasmic localization in LGN KO cells. (**B**) E16.5 epidermal basal cell transduced with mRFP1-AGS3 (magenta) and stained for LGN (green). RFP signal shows

*Figure 2 continued on next page*

*Figure 2 continued*

AGS3 cytoplasmic localization. (**C**) Fluorescence intensity quantification of whole-cell RFP signal in mRFP1-AGS3 transduced basal cells. Overexpression of mRFP1-AGS3 primarily localizes cytoplasmically. (**D**) E16.5 epidermal basal cell transduced with AGS3-V5 and stained for V5 (magenta). (**E**) Image from a movie of a *Krt14^Cre^; Rosa26^mT/mG^* E16.5 epidermis showing membrane GFP (green), transduced with AGS3-mKate2 (magenta). At bottom, cropped images from yellow dashed box area showing unprocessed (raw) image (left), and denoised/bleach corrected image (right). (**F**) Denoised stills from movie in (**E**); *t* = 0 represents metaphase–anaphase transition. Scale bars: 10 µM (**A, B, D, F**), 50 µm (**E**). Here and in all subsequent figures: dashed white (basement membrane), and dashed yellow line (rough outline of cell borders).

The online version of this article includes the following figure supplement(s) for figure 2:

**Figure supplement 1.** AGS3 is mainly cytoplasmic throughout mitosis.

To circumvent this possibility, we transduced the epidermis mosaically (~50%) with our second generation C-terminally tagged AGS3-mKate2 lentiviral construct, and characterized division angles in Survivin+ late-stage mitotic cells, using a pan-TagRFP antibody to discriminate between mKate2+ and mKate2– cells (*Figure 3C*). Comparisons of cumulative frequency distributions of division angles revealed a significant upward shift in the mKate2+ population relative to mKate2– cells (*Figure 3D*), marked by a higher frequency of planar divisions (64 vs. 47%) and lower frequency of perpendicular

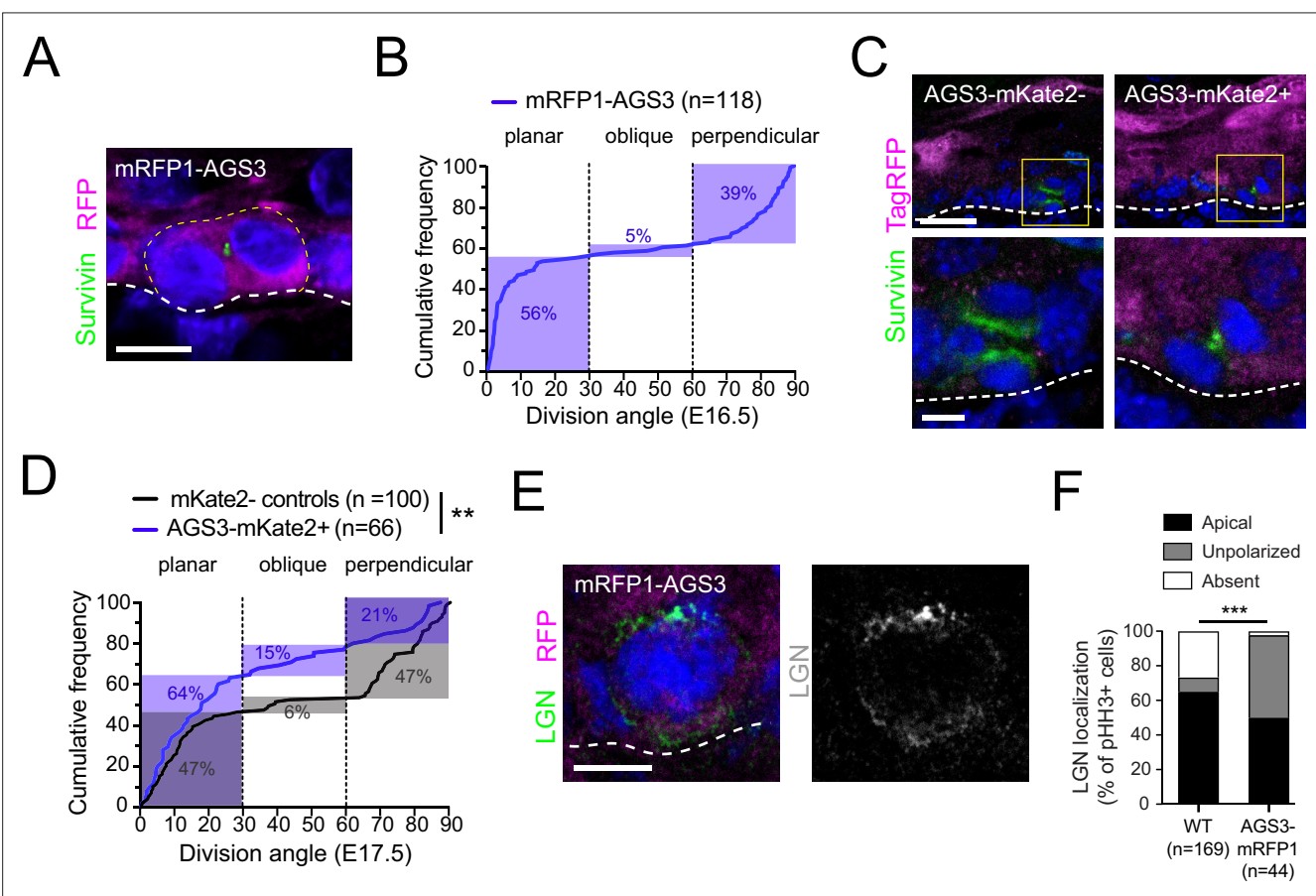

**Figure 3.** AGS3 overexpression promotes planar divisions and disrupts LGN localization. (**A**) Image of telophase basal cell from E16.5 epidermis transduced with lentiviral N-terminal tagged mRFP1-AGS3. (**B**) Cumulative frequency distribution of division angles for mRFP1-AGS3+ basal cells. (**C**) Images of telophase cells from E16.5 epidermis transduced with C-terminal tagged AGS3-mKate2, showing examples of mKate2– (left) and mKate2+ (right) cells. Survivin (green) was used to detect late-stage mitotic cells while a TagRFP antibody (magenta, Invitrogen R10367) was used to detect mKate2, and to discriminate between positive and negative cells. Boxed insets are shown at higher magnification in lower images. (**D**) Cumulative frequency distribution plots of division angles for mKate2+ (blue) and mKate2– (black) cells from mosaically transduced epidermis. Note that mKate2+ cells express AGS3-mKate2 as well as the *Gpsm1^1147^* shRNA to knock down endogenous AGS3, so that exogenous AGS3-mKate2 should be the majority species. (**E**) Example of 'unpolarized' LGN expression in a mitotic basal cell-overexpressing mRFP1-AGS3. (**F**) Quantification of LGN expression patterns observed in wild-type (WT) and mRFP1-AGS3+ mitotic cells. Scale bars: 5 µm in A, C (bottom), E; 20 µm in C (top). **p < 0.01 by Kolmogorov–Smirnov test (**D**); ***p < 0.001 by chi-square test (**F**).

divisions (21 vs. 47%). Of note, this distribution is very similar to what was observed in LGN KOs (*Figure 1C*), suggesting that misexpression of AGS3 could lead to reduction or loss of LGN's spindle orienting function. Consistent with this possibility, overexpression of mRFP1-AGS3 led to a change in the localization pattern of LGN in phospho-histone H3+ (pHH3+) mitotic cells (*Figure 3E*). In WT basal cells, LGN is asymmetrically apically polarized in ~60% of mitoses; however, when AGS3 is overexpressed, this proportion is reduced to ~40%, and LGN more frequently becomes unpolarized (*Figure 3F*).

## AGS3/Gpsm1 loss enhances the apical localization of LGN/Gpsm2

Since overexpression of AGS3 seemed to disrupt the ability of LGN to polarize, we hypothesized that removing AGS3 should have the opposite effect, and perhaps stabilize apical LGN. To test this, we first examined LGN localization in pHH3+ basal cells in E16.5 AGS3 KDs and WT littermate controls (*Figure 4A*). As expected, loss of AGS3 resulted in a significant increase in the proportion of mitotic basal cells with apically recruited LGN, and reduction in the proportion where LGN was not detectable (*Figure 4B*).

While this qualitative assessment revealed that AGS3 loss promoted LGN polarization, we next sought to measure whether the amount of cortical LGN was affected by AGS3 loss. To this end, we quantified the fluorescent intensity (F.I.) of LGN at the perimeter of the cell cortex, as defined by E-cadherin. From this linescan data, we could extract information about LGN maximum F.I., cortical coverage, orientation of the crescent, and integrated F.I. (area under the curve, AUC). An example of a WT mitotic cell with these parameters defined is shown in *Figure 4C*. To minimize potential variability in immunostaining conditions, we compared measurements of LGN F.I. for RFP+ (AGS3 KD) and RFP− (WT internal controls) within the same transduced embryos (representative examples shown in *Figure 4D, E*). Cumulative intensity plots for WT (n = 46) and AGS3 KD (n = 70) cells revealed that the average LGN peak intensity was larger when AGS3 is lost (*Figure 4F*). Raw data for each individual cell measurement are depicted in *Figure 4—figure supplement 1A, B*. LGN maximum F.I., integrated F.I., and cortical coverage were all significantly greater in RFP+ cells (*Figure 4G–I*). However, the apical positioning of the cortex was similar in both RFP+ and RFP− cells (*Figure 4J*). Together these data indicate that AGS3 loss leads to (1) a higher proportion of mitotic cells that polarize LGN, and (2) an increase in the amount of LGN that is present at the apical cortex.

## LGN/Gpsm2 and AGS3/Gpsm1 play opposing roles during telophase reorientation

Previously, we made the surprising discovery that a significant portion of divisions show oblique (30°–60°) orientations at anaphase onset, but later resolve to planar or perpendicular during a process we call telophase correction (*Lough et al., 2019*). In WT cells, roughly equal proportions of oblique divisions correct to planar and perpendicular, so any deviation to this ratio suggests an error in reorientation bias. We noted that, as in pHH3+ early stage mitotic cells, AGS3 KD basal cells also showed abnormal, and frequently persistent, LGN expression during telophase (*Figure 4—figure supplement 1C, D*). This finding prompted us to investigate division dynamics in WT, AGS3 KO, and LGN KO epidermis using ex vivo live imaging. To achieve fluorescent labeling of cell membranes in the epidermis, we bred AGS3 and LGN KO mice onto the $Krt14^{Cre}$; $Rosa26^{mTmG}$ background, performed confocal imaging on E16.5 epidermal explants at 5-min intervals, and used N2V to enhance the membrane-GFP (mG) signal (*Figure 5A*).

An example of a WT basal cell undergoing a perpendicular division, shown in both the native *en face* (*xy*) and z-projection (*xz*) views can be seen in *Figure 5B*. Examples of representative LGN KO and AGS3 KO cells are shown in *Figure 5C, D*, while additional examples of movie stills in different genotypes can be found in *Figure 5—figure supplement 1A–D*. In agreement with our previous studies (*Lough et al., 2019*)—conducted on a different genetic background with deconvolution postprocessing rather than denoising—as a population, WT cells displayed 'randomized' division orientation at anaphase onset but corrected to a bimodal/inverse sigmoidal profile by 1 hr later (*Figure 5E*, black lines).

Our previous data showed that LGN knockdown led to an increase in the percentage of cells that enter anaphase with planar spindles (*Lough et al., 2019*), which was expected because apical LGN promotes perpendicularly oriented spindles during early mitosis (*Williams et al., 2011*). While LGN

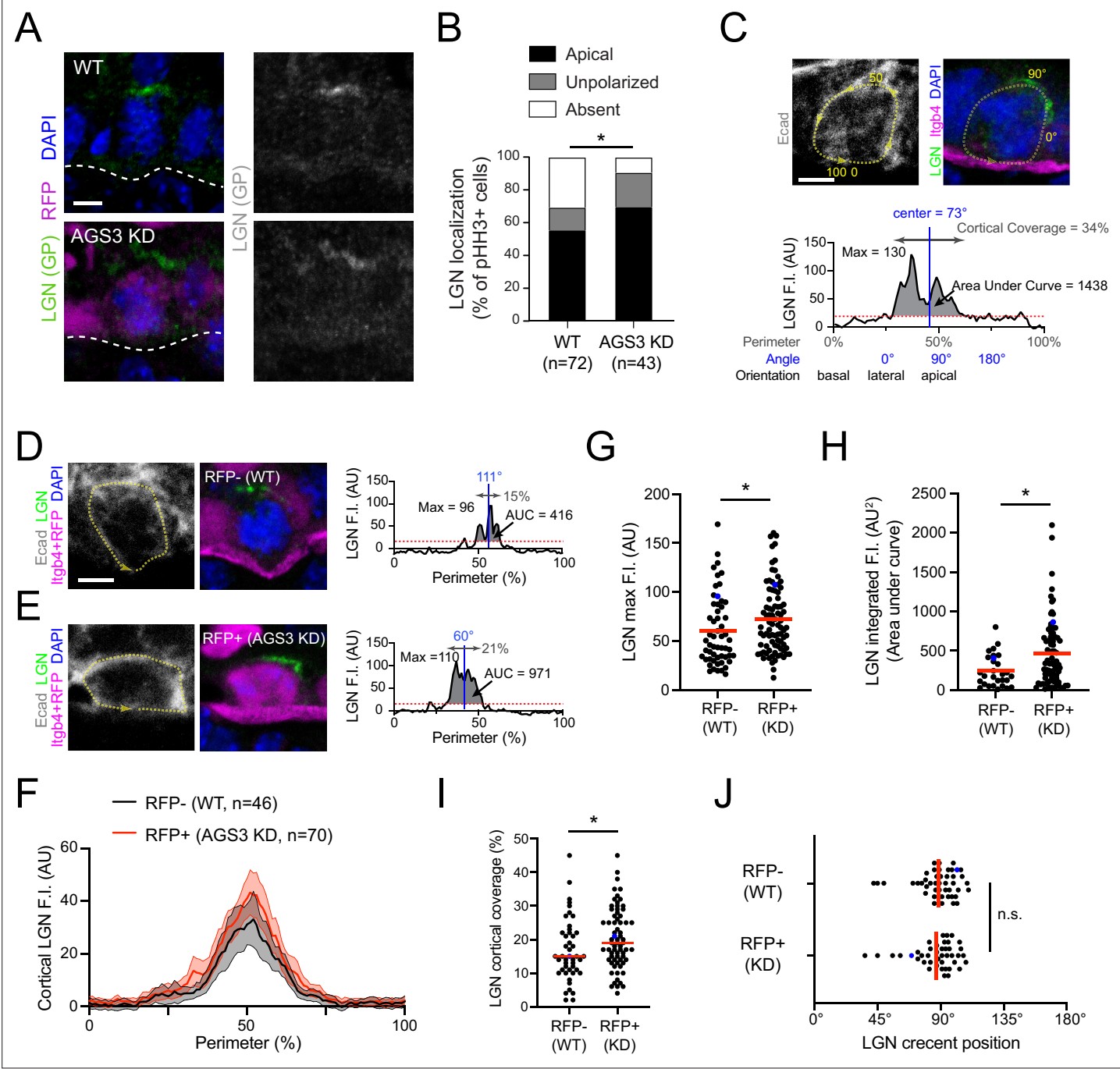

**Figure 4.** AGS3 loss enhances cortical localization of LGN. (**A**) Images of LGN (green) and H2B-RFP (magenta) antibody staining from E16.5 sagittal sections of wild-type (WT; top) and AGS3 KD (bottom) prophase basal cells. (**B**) Quantification of LGN localization patterns in WT (RFP−) and AGS3 KD (RFP+) pHH3+ mitotic basal cells. (**C–J**) Quantification of LGN cortical fluorescent intensity (F.I.) in RFP− controls and RFP+ AGS3 KD mitotic cells. A visual depiction of the methodology used for an example cell is shown in (**C**), with representative RFP− and RFP+ cells shown in (**D, E**). E-cadherin staining is used to define the cell perimeter (0–100% starting from basal; yellow dashed line), with linescans along this mask used to measure LGN F.I., as shown in the graphs in (**C–E**). Red dashed lines indicate the threshold (20 AU) used to discriminate signal from noise. (**F**) Aggregate graphs depicting the mean F.I. (±95 CI) by position for RFP− (*n* = 46) and RFP+ (*n* = 70) cells. (**G**) LGN maximum F.I., which equals the highest value of the cortical linescan for each cell. (**H**) Integrated F.I., calculated as the area under this curve (AUC, shaded regions in C–E). (**I**) Cortical coverage, calculated as the contiguous proportion of the perimeter where the signal exceeds this threshold (gray double arrows in C–E). (**J**) LGN crescent position, defined as the center of this area of cortical coverage (depicted by blue vertical lines in C–E). Blue dots in (**G–J**) represent the actual cells shown in (**D, E**). Scale bars, 5 µm. n.s., not significant; *p < 0.05 by chi-square (**B**) or Mann–Whitney test (**G–J**).

*Figure 4 continued on next page*

*Figure 4 continued*

The online version of this article includes the following figure supplement(s) for figure 4:

**Figure supplement 1.** LGN localization is altered upon AGS3 loss.

knockdown cells corrected exclusively toward planar, we were unable to make any definitive conclusions about a 'maintenance' role for LGN during telophase reorientation because of the small number of cells which entered anaphase at >30°. Here, we address this question by collecting a larger live-imaging dataset of LGN KO epidermal explants, where denoising also enhanced our ability to detect the mG membrane signal and measure division angles in z-projections with precision.

Compared to WT, LGN KO basal cells showed a strong bias toward planar and oblique orientations at anaphase onset, which was significantly accentuated 1 hr later (*Figure 5E*, red lines). At the individual cell level, WT cells entered anaphase at planar (0°–30°, 41%), oblique (30°–60°, 26%), and perpendicular (60°–90°, 33%) angles in roughly equal proportions (*Figure 5F*). By comparison, in LGN KO mutant explants, 67% of cells entered anaphase at planar orientations, confirming that genetic loss of LGN leads to a similar increase in planar-oriented spindles as we observed upon LGN KD using the *Gpsm2*[1617] shRNA. Of note, among the obliquely oriented spindles, correction was equally likely in either direction in WT explants, while in LGN KO explants, >85% (12/14) corrected to planar (*Figure 5G*, example shown in *Figure 5C*). These data strongly suggest that apical LGN not only directs initial spindle positioning during early mitosis, but also promotes perpendicular reorientation during late mitosis.

Next, we performed the same experiments on an AGS3 KO background to determine whether AGS3 regulates initial spindle positioning, reorientation, or both. Once again, WT littermate controls refined from an evenly distributed to a bimodal pattern during the anaphase–telophase transition period (*Figure 5H*, black lines). At the population level, the distribution of division angles in AGS3 KOs displayed a downward shift—indicating a perpendicular bias compared to WT littermates (*Figure 5H*). At the individual cell level, WT obliques were once again similarly likely to correct to planar or perpendicular. However, among AGS3 KO obliques, the vast majority (72%, 13/18) corrected to perpendicular (*Figure 5I, J*; example shown in *Figure 5D*). These data demonstrate that while LGN influences both initial spindle positioning and reorientation, the effect of AGS3 is more pronounced in telophase correction. Moreover, while LGN promotes perpendicular reorientation, AGS3 promotes planar reorientation.

### *Gpsm2* (LGN) is epistatic to *Gpsm1* (AGS3)

We have shown that AGS3 overexpression or loss can inhibit or enhance, respectively, the apical localization of LGN, and that AGS3 and LGN have opposing effects on telophase correction during oriented cell divisions. If AGS3 inhibits the activity of LGN, then LGN would act downstream of AGS3 and we would predict that *Gpsm1* loss should not affect the *Gpsm2* mutant phenotype. To test this, we compared spindle orientation phenotypes—in fixed tissue and using ex vivo live imaging—caused by dual loss of LGN and AGS3 to loss of LGN alone.

We generated two dual loss-of-function models for AGS3 and LGN. First, we used our in utero lentiviral transduction to generate mosaic *Gpsm1*[1147] knockdown on a *Gpsm2*[−/−] null background and compared RFP+ (LGN KO + AGS3 KD) to RFP− (LGN KO − AGS3 KD) populations. As an alternative, we interbred the *Gpsm1* and *Gpsm2* lines to create germline double KOs (AGS3 + LGN dKO). Cumulative frequency histograms of Survivin+ terminal-stage mitotic cells revealed strong biases toward planar divisions in all groups, with no significant differences between them (*Figure 6A, B*). Thus, loss of AGS3 has no effect on the LGN phenotype of increased planar divisions in fixed tissue.

Next, we crossed *Gpsm1* and *Gpsm2* alleles onto the *Krt14*[Cre]; *Rosa26*[mTmG] background to perform ex vivo live imaging. Due to complex breeding schemes and the six alleles needed to generate these mice—as well as the small litter sizes obtained from AGS3 KOs—we were not able to obtain LGN single KOs from the same litters. Thus, we compare the double mutants imaged in this experiment to the same LGN KO group shown in *Figure 5C–E*. On cumulative frequency histograms, both LGN KO and AGS3 + LGN dKO populations showed a similar significant 'leftward' shift toward increasing planar divisions 1 hr after anaphase onset, with nearly overlapping curves (*Figure 6C*). At the individual cell level, cells were twice as likely to enter anaphase at planar compared to non-planar orientations for both LGN KOs and AGS3 + LGN dKOs, and correction was almost universally planar in both

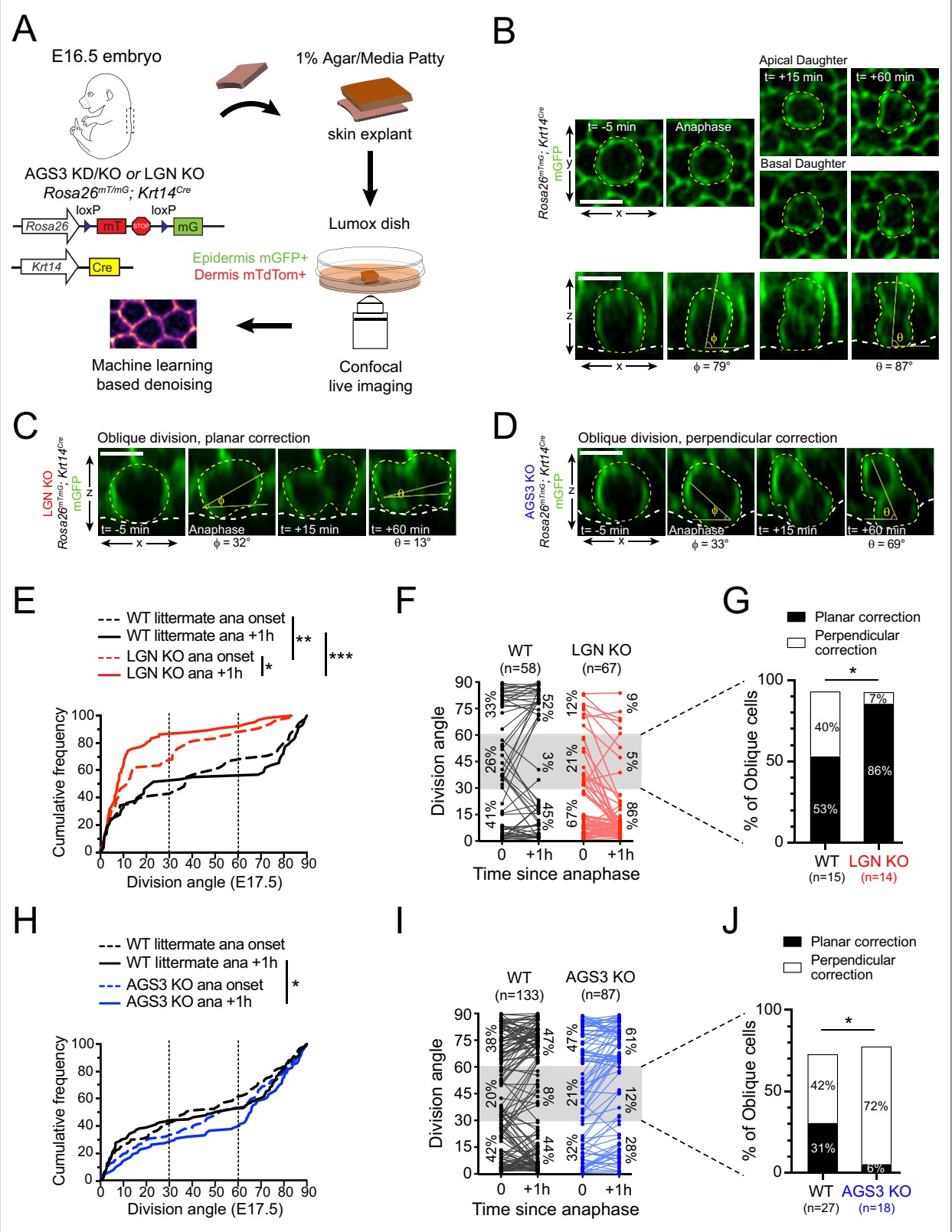

**Figure 5.** AGS3/Gpsm1 loss biases telophase reorientation toward perpendicular. (**A**) Schematic for ex vivo live imaging of wild-type (WT), AGS3 KO (*Gpsm1⁻/⁻*), and LGN KO (*Gpsm2⁻/⁻*) embryonic epidermal explants on a *Krt14^Cre^; Rosa26^mT/mG^* background, where epithelial cell membranes are GFP+. (**B**) Native *en face* (top) and z-projections (bottom) movie stills of a WT mitotic cells as it enters anaphase (*t* = 0), through 1 hr later, depicting a perpendicular division. Division orientation angles are shown below (φ, anaphase onset; θ, +1 hr). Z-projection movie still from equivalent timepoints

*Figure 5 continued on next page*

*Figure 5 continued*

showing a LGN KO (**C**) or AGS3 KO (**D**) division. See *Figure 4—figure supplement 1C, D* for additional timepoints. (**E**) Cumulative frequency distribution of division orientation for E17.5 LGN KO embryos at anaphase onset and +1 hr later. (**F**) Line graphs of individual cell data from (**E**) depicting orientation at anaphase onset and 1 hr later for LGN KO cells (red) and WT littermate controls (black). Percentages of cells in each orientation bin are shown to the left ($t = 0$) and right ($t = +1$ hr) of the data points. (**G**) Data from (**F**) depicting behavior of anaphase ($t = 0$) obliques (gray zone in (**E**)), and frequency of planar vs. perpendicular correction; rare cells that remain oblique are not included. (**H–I**) Similar plots as (**E–G**) for AGS3 KO cells (blue) and WT littermate controls (black). Scale bars, 10 µm; *n* values indicate events from three to four embryos imaged in two technical replicates per genotype; *p < 0.05, **p < 0.01, ***p < 0.001 by Kolmogorov–Smirnov test (**E, H**) or Fisher's exact test (**G, J**).

The online version of this article includes the following figure supplement(s) for figure 5:

**Figure supplement 1.** Additional movie stills of telophase correction behavior.

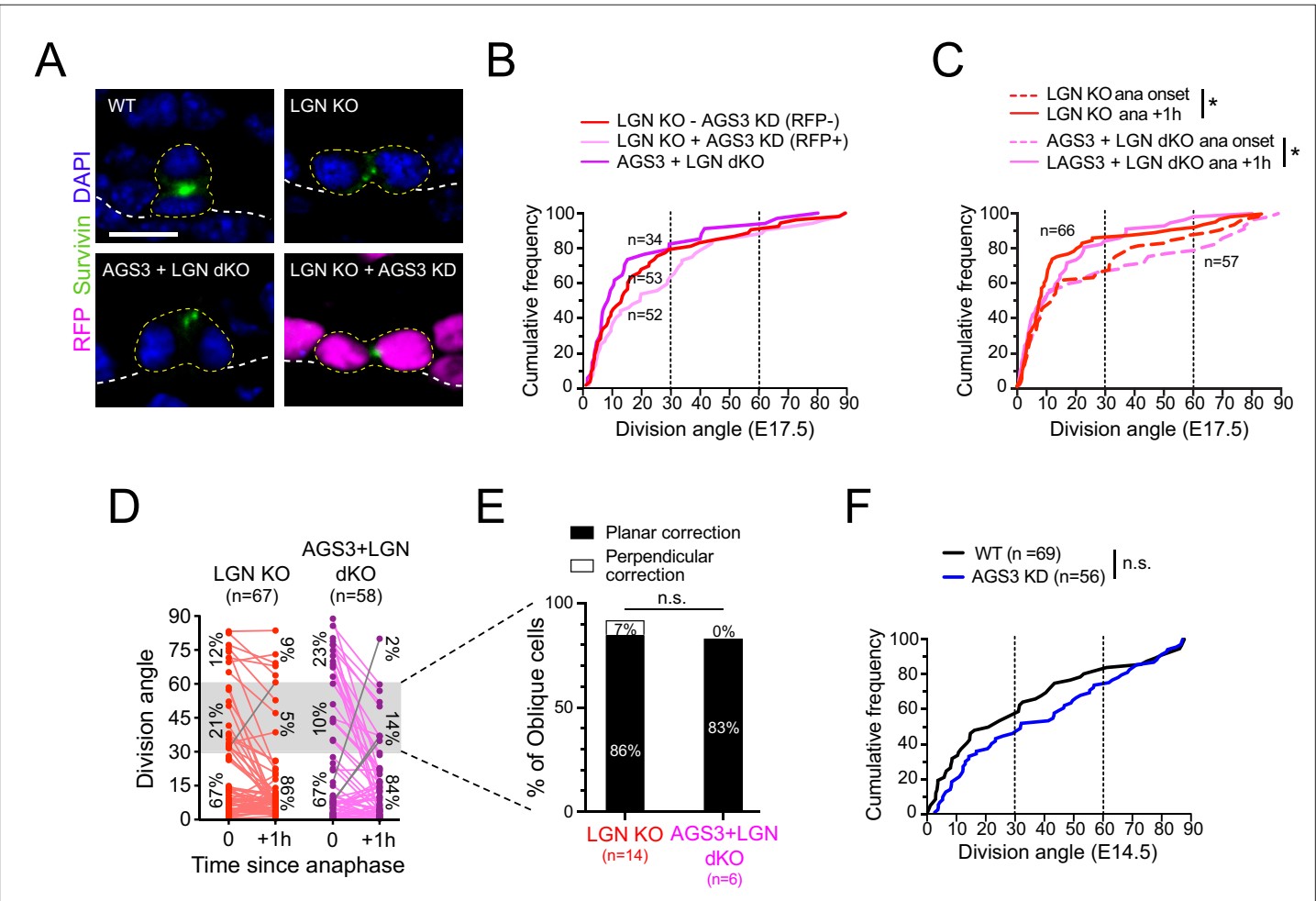

**Figure 6.** *Gpsm2* is epistatic to *Gpsm1*. (**A**) Images of telophase cells from E17.5 wild-type (WT), LGN KO (*Gpsm2⁻/⁻*), AGS3 + LGN dKO (*Gpsm1⁻/⁻; Gpsm2⁻/⁻*), and AGS3 KD + LGN KO (*Gpsm1¹¹⁴⁷; Gpsm2⁻/⁻*) backskin epidermis, labeled with the late-stage mitotic marker Survivin (green). (**B**) Cumulative frequency distribution of telophase division angles from fixed sections for indicated genotypes: *Gpsm2⁻/⁻; Gpsm1¹¹⁴⁷* H2B-RFP– (LGN KO, red dashed line), *Gpsm2⁻/⁻; Gpsm1¹¹⁴⁷* H2B-RFP+ (LGN KO +AGS3 KD, magenta dashed line), and *Gpsm1⁻/⁻; Gpsm2⁻/⁻* (AGS3 + LGN dKO, magenta line). (**C**) Cumulative frequency distribution of division angles at anaphase onset (dashed lines) and 1 hr later (solid lines) from live-imaging experiments of LGN single KOs (red, replotted from *Figure 4E*) and AGS3 + LGN dKOs (magenta). (**D**) Line graphs of individual cell data from (**C**) depicting orientation at anaphase onset and 1 hr later for LGN KO (red) and AGS3 + LGN dKO cells (magenta). (**E**) Data from (**D**) depicting planar vs. perpendicular telophase correction behavior for anaphase obliques (gray bar), showing that 83% of double knockout cells correct to planar, similar to the 86% observed in LGN single KOs. (**F**) Cumulative frequency distribution of telophase division angles quantified from E14.5 back skin sections for AGS3 KD (*Gpsm1¹¹⁴⁷* H2B-RFP+) cells (blue) and WT uninjected littermate controls (black). Scale bars, 10 µm; *n* value indicates cells from >3 independent embryos per genotype (**B**) with at least 2 technical replicates (**C–E**); n.s., not significant; *p < 0.05, by Kolmogorov–Smirnov test (**B, C, F**) or chi-square test (**E**).

groups (exceptions shown as gray lines in *Figure 6D*). Finally, among anaphase obliques, over 80% in both genotypes corrected to planar (*Figure 6E*). These data confirm that *Gpsm2* is epistatic to *Gpsm1* in telophase correction.

Finally, we reasoned that if AGS3 required LGN for its function, then AGS3 loss should have no effect when LGN is dispensable. To test this, we took advantage of the fact that epidermal stratification occurs in both LGN-independent and -dependent phases. In the single-layered epithelium (E12.5 and earlier), divisions are initially planar, become 'randomized' by E13.5–E14.5 when stratification commences, and finally adopt their mature 'bimodal' distribution by ~E15.5–E16.5 (*Damen et al., 2021*; *Lechler and Fuchs, 2005*; *Williams et al., 2014*). LGN first shows apical polarization at ~E15.5, and LGN loss has no effect on spindle orientation prior to this age (*Williams et al., 2014*). Thus, the initial phases of stratification occur independently of LGN. Interestingly, like LGN, loss of AGS3 in early (E14.5) epithelia caused no significant effect on spindle orientation (*Figure 6F*). Collectively, these data show that AGS3 loss has no apparent effect when LGN is absent, and strongly suggest that AGS3 acts through LGN in spindle orientation.

## AGS3/Gpsm1 loss promotes asymmetric cell fates and differentiation

Spindle orientation is frequently, but not always linked to cell fate choices (*Williams and Fuchs, 2013*). Correlative studies have shown that loss of core apical complex proteins such as Gαi3, Insc, Par3, NuMA, and LGN—which reduce perpendicular divisions—also lead to epidermal thinning, thus demonstrating that they impact stratification (*Seldin et al., 2016*; *Williams et al., 2011*; *Williams et al., 2014*). As a first measure of the potential impact of AGS3 or LGN loss on epidermal architecture, we quantified the thickness of the Krt10+ spinous and granular layers as a measure of differentiation (*Figure 7A, B*). No changes in spinous thickness were observed in either mutant at E15.5, consistent with the lack of any effect on spindle orientation by either gene at this age. However, by E17.5 we observed a significant decrease in spinous thickness in LGN KOs compared to WT littermates (*Figure 7C*), consistent with previous observations in LGN KDs (*Williams et al., 2011*). While the effect was milder in AGS3 KOs, we also observed a significant increase in spinous thickness at E17.5 (*Figure 7D*). Thus, during epidermal development, AGS3 loss leads to increased differentiation.

Our ex vivo live-imaging studies have revealed previously unappreciated plasticity in daughter cell positioning during late stages of mitosis. Thus, it is possible that daughter cells could either differentiate following divisions or alternatively, reintegrate into the basal layer following mitosis. Such behaviors have been recently documented in other epithelia (*McKinley et al., 2018*; *Wilson and Bergstralh, 2017*), and dedifferentiation has been reported following wounding in adult skin (*Donati et al., 2017*). Due to technical challenges of following daughter cell fates via long-term ex vivo imaging of embryonic explants, we instead relied on clonal analyses in fixed tissues to determine the effect of AGS3 and LGN loss on cell fate choices.

Genetic lineage tracing is a powerful tool to identify clonally related cells, and relies on induction of a permanent genetic mark—usually a fluorescent protein—by a tissue-specific, inducible Cre recombinase (*Figure 7E*). Previously, we and others have used short-term 'mitotic' genetic lineage tracing to characterize self-renewing and differentiating behaviors (*Byrd et al., 2019*; *Poulson and Lechler, 2010*; *Williams et al., 2014*). These include the following clone types: (1) symmetric cell divisions (SCDs), which contain two basal cells; (2) ACDs, which contain one basal and one suprabasal cells; and (3) delamination events, which contain a single suprabasal (spinous) cell. Across these studies, there has been a good correlation between the proportions of SCD and ACD clones with frequencies of planar and perpendicular divisions in fixed tissue. However, whether loss of essential spindle orientation genes affects cell fate choices has never been directly tested.

Recently, we used mitotic genetic lineage tracing to show that loss of the adherens junction protein afadin (*Afdn*), which interferes with telophase reorientation fidelity, leads to an increase in asymmetric fate clones (*Lough et al., 2019*). Here, we apply a similar approach using *Krt14*CreERT2; *Rosa26*confetti mice on LGN, AGS3, and AGS3 + LGN dKO backgrounds, treated with a single low dose of tamoxifen at E16.5 and collected at E18.5 (*Figure 7E, F*). Although the *Krt14* promoter is less active in suprabasal cells, in order to minimize inclusion of clones that may have been induced while already differentiated, only single-cell clones in the first spinous layer were counted as delamination events.

We compared the proportion of SCD and ACD clones in AGS3 KOs (*Gpsm1*−/−) and heterozygote/WT (*Gpsm1*+/) littermate controls, and observed a significant shift toward ACDs in KOs (*Figure 7G*).

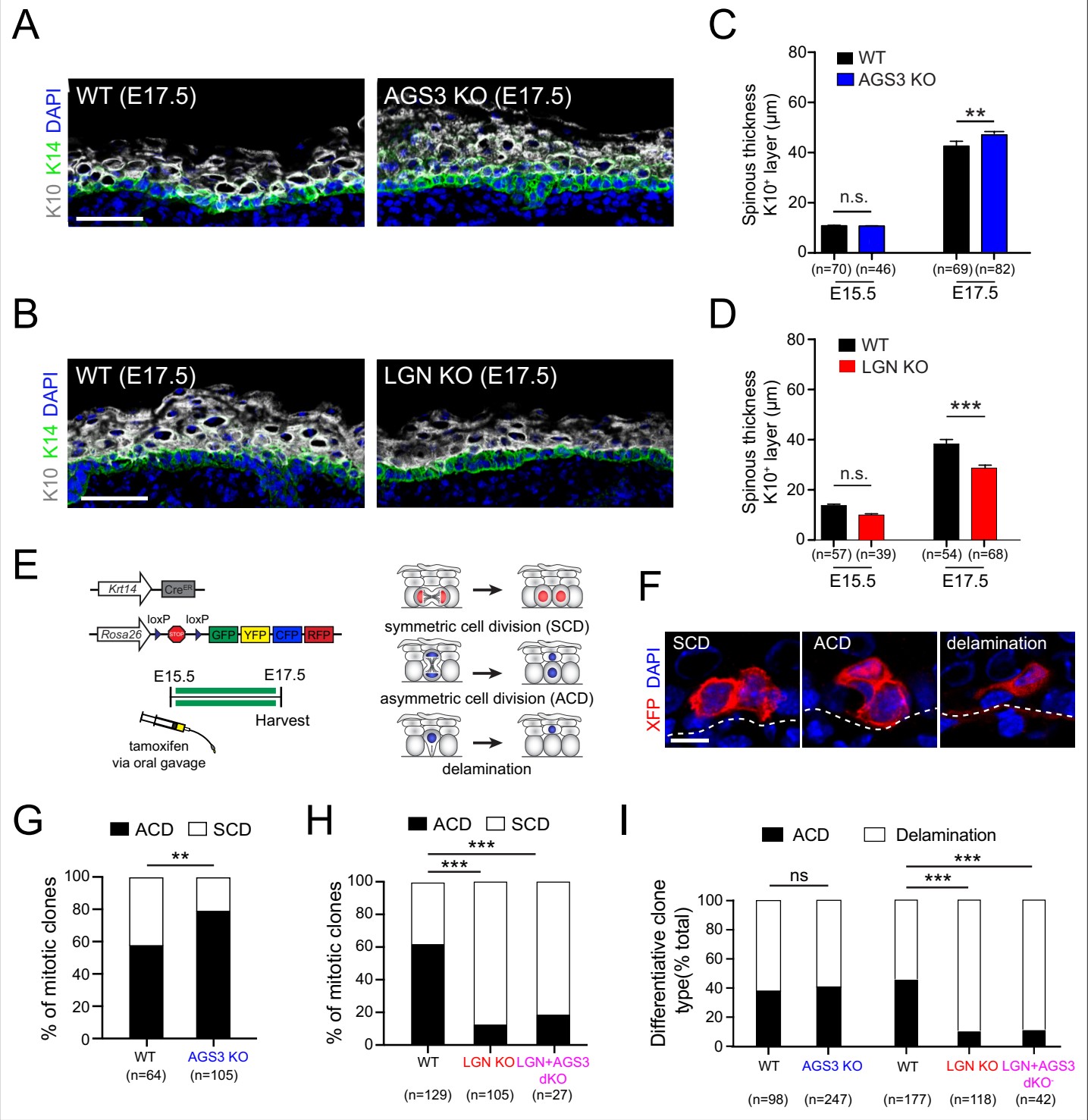

**Figure 7.** AGS3/Gpsm1 loss promotes asymmetric cell fates and differentiation. Immunofluorescent images (**A, B**) and quantification (**C, D**) of spinous (K10, gray) layer thickness in AGS3 KOs (**A, C**) and LGN KOs (**B, D**) compared to wild-type (WT) littermate controls. (**E**) Graphical depiction of clonal lineage tracing strategy using *Krt14^CreER^* and the multi-colored *Rosa26^confetti^* reporter, for data shown in (**F–I**). (**F**) Representative images of showing three types of clones. (**G**) Quantification of proportion of symmetric cell division (SCD) vs. asymmetric cell division (ACD) clones in AGS3 KOs (*Gpsm1^−/−^*) (blue) compared to heterozygote and WT (*Gpsm1^+/^*) controls (black). (**H**) Same analysis as (**G**) but for WT (black), LGN KO (*Gpsm2^−/−^*, red), and AGS3 + LGN dKO (*Gpsm1^−/−^*; *Gpsm2^−/−^*, magenta) clones. (**I**) Proportions of differentiative clones occurring by ACD vs. delamination for indicated genotypes. Scale bars: 50 μm in (**A, B**), 10 μm in (**F**); *n* values indicate measurements (**C, D**) or clones (**G–I**) from >3 independent embryos per genotype; n.s., not significant, **p < 0.01, ***p < 0.001 by unpaired *t*-test (**C, D**) or chi-square (**G–I**).

On the other hand, loss of LGN led to a sharp increase in SCD clones (*Figure 7H*). In further support of *Gpsm2* being epistatic to *Gpsm1*, AGS3 + LGN dKOs showed a similar increase in SCD clones as LGN KOs (*Figure 7H*). These data confirm that LGN and AGS3 play opposing roles in regulating oriented cell divisions, which result in altered cell fate choices at the clonal level.

It is worth noting that in both AGS3 and LGN mutants, there was agreement between cell fate choices determined by genetic lineage tracing and division orientation at the 'anaphase +1 hr' time-point of our ex vivo live-imaging studies. For example, 74% of AGS3 KO cells showed perpendicular/oblique orientations at +1 hr (*Figure 5I*), and 79% of divisions resulted in ACD clones (*Figure 7G*). Similarly, in LGN KOs, 86% of imaged mitoses adopted planar orientations at +1 hr (*Figure 5F*), and 88% of mitotic clones were SCDs (*Figure 7H*). This suggests that ex vivo live imaging captures the behaviors that occur in vivo, and that measures of telophase orientations accurately reflect fate outcomes.

During epidermal development, differentiation can be accomplished through either ACD or delamination. Delamination is a process by which basal cells initiate differentiation within the basal layer (e.g., via upregulation of spinous keratins such as K10), followed by detachment from the underlying basement membrane and upward migration into the spinous layer (*Cockburn et al., 2022*; *Ellis et al., 2019*; *May et al., 2023*; *Watt and Green, 1982*; *Wickström and Niessen, 2018*; *Williams et al., 2014*). We and others have documented that delamination is the predominant mode of differentiation during early stratification (E12.5–E15.5), while ACDs become more common during peak to late stratification (*Damen et al., 2021*; *Williams et al., 2014*). In both embryonic and adult epidermis, the processes of proliferation and differentiation are spatially correlated and can locally influence each other (*Mesa et al., 2018*; *Wickström and Niessen, 2018*). However, it remains an open question whether mutations that alter the balance between SCDs and ACDs could impact delamination. To test this, we examined whether the proportions of the two types of 'differentiative' clones—ACD and delamination—were affected by LGN or AGS3 loss. While no significant change in differentiation behavior was observed in AGS3 KOs, delamination events were significantly increased in both LGN KOs and AGS3/LGN double KOs, where ACDs are infrequent (*Figure 7I*). These data demonstrate that elevated delamination can partially compensate for the impaired differentiation by ACD observed upon LGN loss, and may explain why the epidermal thinning is less severe than expected in the absence of LGN.

## Discussion

Collectively, these studies show that LGN/Gpsm2 and AGS3/Gpsm1 play opposing roles in regulating oriented cell divisions and fate choices in the developing epidermis (*Figure 8*). Static and live analyses of division orientation, genetic lineage tracing, and quantification of differentiation markers confirm that AGS3 promotes self-renewal through SCDs while LGN promotes differentiation via ACDs. LGN knockdown (*Williams et al., 2011*) or knockout (these studies) leads to a planar bias in division orientation, which we also observe when AGS3 is overexpressed. Conversely, AGS3 loss has the reverse effect, increasing perpendicular divisions. LGN, Pins, and its homologs have evolutionarily conserved roles in determining the axis of the mitotic spindle. Beyond this canonical function, we now provide evidence that LGN is also required during late stages of mitosis to correct 'errant' oblique divisions toward perpendicular. Finally, in the absence of AGS3, we observe enhanced cortical LGN localization throughout mitosis and demonstrate that AGS3 promotes planar divisions during telophase correction.

### Mechanisms of competition between LGN and AGS3

In theory AGS3 could promote planar divisions through a variety of mechanisms, to be discussed in greater detail below. However, we provide the following lines of evidence that AGS3 acts as a novel negative regulator of LGN: (1) AGS3 is enriched in the cytoplasm, and thus is unlikely to interact directly with the cortical spindle orientation machinery, (2) AGS3 loss- or gain-of-function promotes or inhibits, respectively, the apical localization of LGN, and (3) *Gpsm2* is epistatic to *Gpsm1*, demonstrating that LGN loss masks AGS3 function, and implying that AGS3 acts through LGN.

In thinking about how GPSM complexes regulate spindle orientation, it is important to consider their accessibility, partner binding affinities, and stoichiometry of their interactions. First, free GPSM

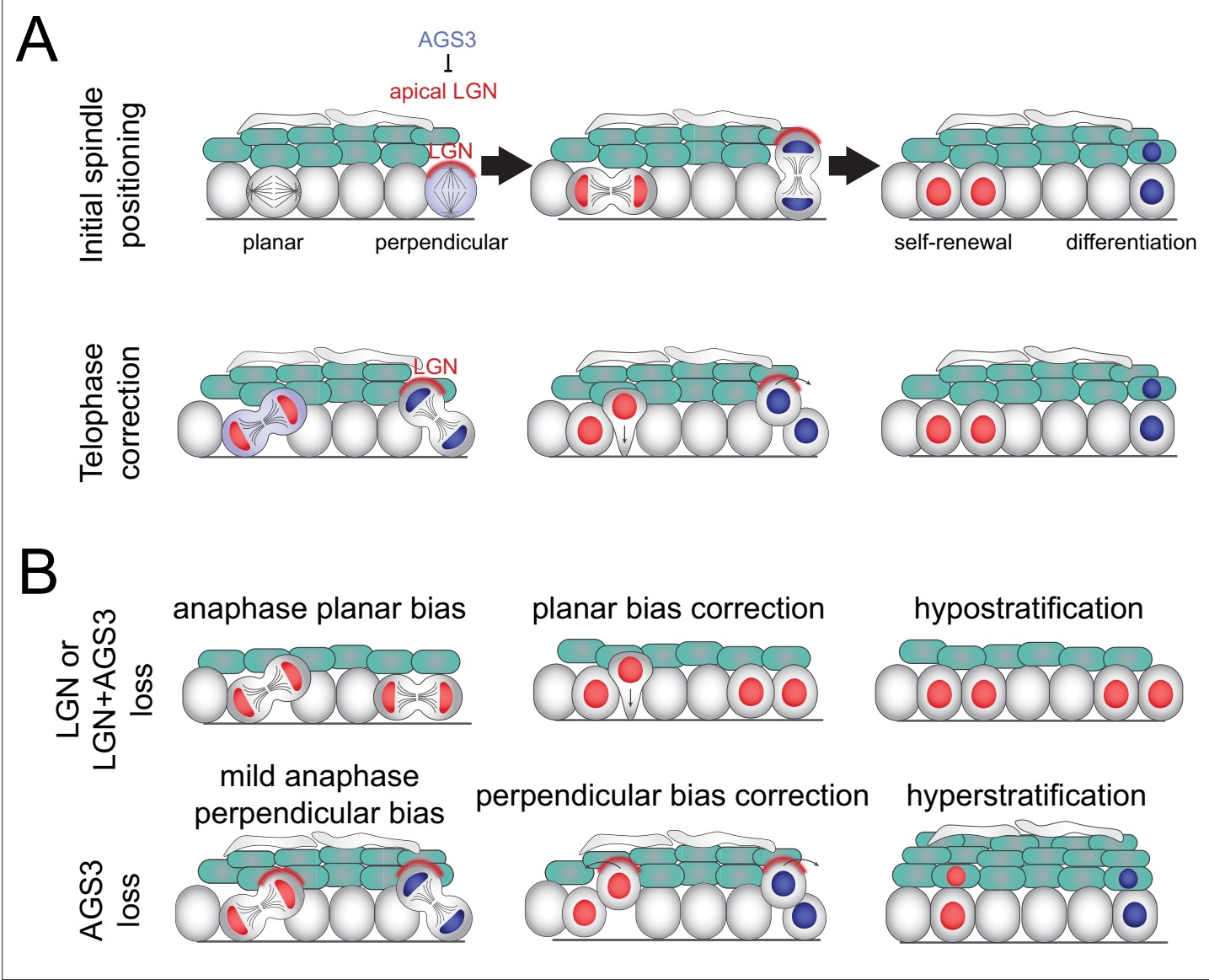

**Figure 8.** Model of two-step process for oriented cell divisions. (**A**) Basal progenitors can undergo either self-renewing planar symmetric divisions (red cell) or differentiative perpendicular asymmetric divisions (blue cell). Oriented cell divisions are regulated in a two-step process: (1) initial spindle positioning (top) and (2) telophase correction (bottom). Polarized apical LGN promotes apical spindles at metaphase and also promotes perpendicular reorientation during telophase. Primarily cytoplasmic AGS3 displaces LGN from the cortex throughout mitosis to inhibit perpendicular divisions. (**B**) Phenotypes resulting from loss of LGN and/or AGS3.

proteins exist in a closed confirmation, where the N-terminal TPR region is bound to the C-terminal GPR repeats. The only protein known to be capable of binding LGN in its closed conformation is Gαi—preferentially in its GDP-loaded form—which is able to access the first GoLoco motif that is not involved in intramolecular interactions (***Du and Macara, 2004***; ***Nipper et al., 2007***; ***Takayanagi et al., 2019***). Thus, Gαi-GDP binding is thought to be an early event in complex formation, promoting cortical association of GPSM proteins due to Gαi myristoylation, and catalyzing additional Gαi binding that promotes the open conformation. An important corollary of the Gαi-GDP preference of GPR domains is that GPSM proteins act as guanine dissociation inhibitors that reduce the exchange of GDP for GTP on Gαi subunits, which in some contexts can be opposed by the guanine nucleotide exchange factor Ric-8A (***David et al., 2005***; ***Hampoelz et al., 2005***; ***Woodard et al., 2010***).

Second, structural studies have shown that LGN TPR domains cannot simultaneously bind multiple proteins, for example NuMA and Insc (***Culurgioni et al., 2011***; ***Yuzawa et al., 2011***; ***Zhu et al.,***

*2011b*). While an early model posited that Insc recruits LGN to the apical cortex and then hands it off to NuMA, the very high affinity of Insc for LGN suggests that LGN–Insc and LGN–NuMA are more likely to exist in separate pools (*Culurgioni and Mapelli, 2013*; *Culurgioni et al., 2018*). Nonetheless, it remains possible that post-translational modifications could weaken this interaction and facilitate the dissociation of LGN from Insc.

Third, NuMA can bind both microtubules and dynein directly and is thought to be anchored by the LGN–NuMA–Gαi ternary complex at the plasma membrane to ensure correct spindle positioning (*Kotak et al., 2012*; *Seldin et al., 2016*). Recent structural studies have shown that LGN and NuMA can exist in a heterohexameric ring structure, which forms higher order networks that facilitate microtubule capture and orient spindles in mammalian cells (*Pirovano et al., 2019*). It is not presently known whether AGS3 and NuMA can also form these higher order oligomers, but as discussed further below, key residues of LGN that mediate this interaction are missing in AGS3.

At a molecular level, we favor a model in which AGS3 competes with LGN for NuMA binding, preventing LGN from forming productive ternary complexes that capture astral microtubules at the apical cortex. This hypothesis is based on several lines of evidence. First, there is direct biochemical evidence that AGS3 can bind to NuMA (*Saadaoui et al., 2017*), and the specific residues within the TPR5/6 domains of LGN that mediate binding to NuMA—N203, R221, and R236 (*Culurgioni et al., 2011*; *Zhu et al., 2011b*)—are conserved in AGS3. Second, a construct consisting of the AGS3-TPR domains localizes to spindle poles—where NuMA is present—while the TPR domains of LGN localize to the cytoplasm (*Saadaoui et al., 2017*). Moreover, an LGN chimera containing the AGS3 TPR domain can displace WT LGN from the cell cortex but can only partially rescue a *Gpsm2* spindle orientation defect (*Saadaoui et al., 2017*). This suggests that the TPR domains of AGS3 might have a higher affinity for NuMA than LGN, but cannot orient spindles efficiently, perhaps because AGS3-NuMA cannot form a functional apical complex. Third, mutations in LGN, or NuMA, that render them oligomerization deficient—for example, unable to form hexameric ring structures—lack spindle orienting ability (*Pirovano et al., 2019*). Thus, the ability of GPSM protein TPR domains to form higher-order complexes with NuMA appears to be critical for their function.

These studies also showed that the curvature of the NuMA–LGN hexamer is accomplished by the unusually long fourth TPR domain of LGN—which contains 54 amino acids instead of 34 (*Pirovano et al., 2019*). Interestingly, murine AGS3 shows the poorest conservation with LGN in this particular domain (63% similar compared to >86% similarity for all other TPRs), with the highest dissimilarity is in the region between the two alpha-helical regions (data not shown). While the crystal structure of the AGS3 TPR has not been solved, the divergence of this particular TPR that is critical for ring formation suggests that AGS3 may not be capable of forming heterohexamers with NuMA. Moreover, another region of LGN that is critical for its oligomerization capacity is the short N-terminus that precedes the first TPR (*Pirovano et al., 2019*). This N-terminal region is highly conserved among vertebrate *Gpsm2* orthologs, but entirely divergent among *Gpsm1* orthologs (data not shown), providing a second line of evidence that NuMA bound to AGS3 is unlikely to form ring structures.

A caveat of this reductionist model in which LGN and AGS3 play opposing roles in promoting or inhibiting NuMA's cortical recruitment and spindle orienting activity is that while LGN and Pins are required for NuMA and Mud (the *Drosophila* ortholog of NuMA) to localize to the cell cortex, they are not always sufficient. For example, the existence of an LGN-independent mechanism to recruit NuMA to the apical and/or basal cortex could explain the small proportion of cells that are still able to execute perpendicular divisions in LGN KOs. Studies of HeLa and keratinocyte cells in culture have shown that LGN is required for NuMA's cortical recruitment through metaphase, but dispensable at anaphase, where instead, 4.1 G/R serves to anchor NuMA to the cortex (*Kiyomitsu and Cheeseman, 2013*; *Seldin et al., 2013*). Similarly, a recent study in *Drosophila* follicular epithelium showed that Pins is recruited apically and laterally during mitosis, while Mud is lateral (*Neville et al., 2023*).

This same study also showed that it is not the localization of Pins, but its upstream binding partner Insc, which is the best predictor of spindle orientation in *Drosophila* follicular epithelium, and raised the intriguing idea that there is a threshold level of Insc expression, above which it localizes apically—causing spindle misorientation—and below which it localizes laterally (*Neville et al., 2023*). Since AGS3 can also bind Insc (*Izaki et al., 2006*; *Yuzawa et al., 2011*), we also consider the possibility that cytoplasmic AGS3 may sequester Insc from the apical cortex in epidermal basal cells. A prediction of this model is that reduction of AGS3 would elevate Insc levels, while overexpression of AGS3 would

reduce Insc levels. Following the Bergstralh threshold model, high levels of Insc could lead to its apical polarization while lower levels may lead to lateral or unpolarized Insc, with LGN presumably following Insc's lead. While we are unable to observe endogenous Insc due to a lack of suitable antibodies, the observation that some AGS3-overexpressing cells show unpolarized rather than apical LGN, is consistent with this Insc threshold model.

While we favor that LGN and AGS3 compete for binding to a common interacting partner such as NuMA or Insc, it is also possible that AGS3 could bind to, and inhibit, LGN directly. For example, it has been shown that the AGS3 GPR region can pull-down the TPR domain of LGN (*Saadaoui et al., 2017*). Unfortunately, crossreactivity between our LGN and AGS3 antibodies makes it difficult to test for colocalization in mitotic basal cells, as even our reasonably specific guinea pig anti-LGN antibody appears to label some AGS3 in LGN KOs (*Figure 1A*). Nonetheless, because AGS3 is primarily cytoplasmic, it could sequester a pool of LGN away from the cell cortex, preventing engagement with NuMA. One caveat to this model is that in order for AGS3 to bind to LGN, they would both need to be in their open conformations, a process believed to require Gαi-GDP binding. Because Gαi proteins are myristoylated and membrane associated, this event would be more likely to occur at the cell cortex.

## Binding partners and subcellular localization of LGN and AGS3

The spindle orienting function of *Gpsm2* and its homologs is highly conserved throughout evolution, and may have evolved as early as the time when bilateria and cnidaria diverged (*Schiller and Bergstralh, 2021*; *Wavreil and Yajima, 2020*). While LGN and its orthologs have an evolutionarily conserved role in oriented cell divisions, the manner and pattern in which LGN localizes to the cell cortex varies among epithelial tissues. For example, in the retina and developing oral epithelia, LGN localizes apically and serves a similar function in promoting perpendicular divisions as in epidermis (*Byrd et al., 2016*; *Lacomme et al., 2016*). On the other hand, in the tactile filiform papilla of the dorsal tongue, radial glia of the subventricular zone, and neural tube, LGN localizes laterally and promotes planar divisions (*Byrd et al., 2016*; *Konno et al., 2008*; *Lacomme et al., 2016*; *Morin et al., 2007*; *Peyre et al., 2011*). Thus, while the spindle orienting capacity of LGN is conserved, its variable subcellular localization influences division directionality differently across tissues.

In addition to Insc and Gαi family proteins, a growing list of proteins—including Dlg, E-cadherin, Frmpd1/4, and afadin—have been found to promote the cortical localization of LGN in different contexts (*Carminati et al., 2016*; *Gloerich et al., 2017*; *Schiller and Bergstralh, 2021*; *Wee et al., 2011*; *Yuzawa et al., 2011*). Although autoinhibition and phosphorylation are mechanisms known to regulate LGN's ability to interact with proteins that promote its membrane association (*Bergstralh et al., 2017*; *Johnston et al., 2009*; *Pan et al., 2013*), comparatively less is known about negative regulators. One example is SAPCD2, which inhibits the cortical localization of LGN in the retina, possibly by competing for NuMA binding (*Chiu et al., 2016*). We believe that AGS3 is another, though clearly its function is context dependent, given its inability to orient spindles or influence LGN localization in neurons (*Saadaoui et al., 2017*).

While unbiased proteomic approaches might best ascertain how complexes containing AGS3 and LGN differ, we speculate that differences in their subcellular localization and function could also be attributable to post-translational modifications such as phosphorylation. The flexible linker domain of LGN/Pins has been shown to be phosphorylated by both aPKC and Aurora kinase, which mediate interactions with 14-3-3 and discs large (Dlg), respectively (*Hao et al., 2010*; *Johnston et al., 2009*). While it is not known whether aPKC phosphorylation of LGN impacts its spindle orienting capacity, loss of *Prkci* in the epidermis does result in a spindle orientation phenotype (*Niessen et al., 2013*). On the other hand, the Dlg–LGN interaction is highly conserved throughout evolution (*Schiller and Bergstralh, 2021*), and a non-phosphorylatable *Drosophila pins*[S436A] mutant is non-functional in an in vitro S2-induced polarity assay and in vivo in follicular epithelial cells (*Hao et al., 2010*; *Neville et al., 2023*). Moreover, the equivalent mutation in vertebrate LGN (S401A) induces spindle misorientation in MDCK cells and the chick neural tube (*Saadaoui et al., 2014*). This serine residue is conserved in the AGS3 linker, but there are some differences in the flanking residues, and it has been reported that the AGS3 linker binds Dlg with ~500-fold lower efficiency than LGN (*Zhu et al., 2011a*). While substitution of the LGN linker into AGS3 is not sufficient to relocalize AGS3 from the cytoplasm to the cortex (*Saadaoui et al., 2017*), the converse experiment has not been attempted. Other potential

phosphorylation sites for LGN include T450, which promotes growth in breast cancer cells (*Fukukawa et al., 2010*).

Another way in which AGS3 and LGN might differ is through their interactions with G proteins. Although LGN and AGS3 bind Gαi proteins with similar affinity in vitro (*Willard et al., 2008*), some of the best insights into how Gαi interactions influence their localization and function in vivo come from domain swap experiments between the GPR regions of LGN and AGS3 (*Saadaoui et al., 2017*). The Morin lab found that while the AGS3 GPR domain cannot substitute for the LGN GPR domain, replacing the inter-GPR regions of AGS3 with those of LGN confers cortical localization. The converse is also true, in that replacing the inter-GPR regions of LGN with AGS3's results in cytoplasmic localization. Moreover, for AGS3 it has been shown that addition of inter-GPR regions significantly enhances Gαi1 binding compared to isolated GPR domains alone (*Adhikari and Sprang, 2003*). Collectively, these findings suggest that the inter-domain regions contain important information that might regulate the ability of LGN/AGS3 to interact with specific Gαi proteins.

Saadaoui et al. speculated that variability in the interdomain regions might differentially affect intramolecular interactions between the TPR and GPR domains of LGN and AGS3, but conceivably, this could also impact their ability to bind specific Gαi proteins, particularly if post-translational modifications occur within these interdomains. There are three Gαi proteins in mammals and all are expressed in the epidermis, but it is Gαi3 which colocalizes with LGN, and *Gnai3* loss leads to a planar bias like loss of LGN (*Williams et al., 2014*). *Gnai2* loss has no obvious phenotype (*Williams et al., 2014*), and *Gnai1* loss has yet to be explored, so it remains possible that AGS3 may preferentially associate with a different cohort of Gαi proteins than LGN in the developing epidermis.

## Are planar divisions an active or 'default' process?

While much remains to be explored at the molecular level of how AGS3 interacts with the spindle orientation machinery, our studies shed new light on both the negative and positive regulation of perpendicular, asymmetric divisions. Importantly, we view AGS3 as a negative regulator of perpendicular divisions, rather than a promoter of planar divisions. We speculate that relative levels of AGS3 and LGN within individual cells determine the likelihood that they will both establish and maintain a perpendicular orientation throughout mitosis. However, this cannot explain why a significant fraction of basal cells 'choose' planar orientations independent of AGS3. The fact that LGN loss—and also combined AGS3/LGN loss—results in a strong planar bias rather than randomization of division angles suggests that either (1) there is an as-yet undiscovered active mechanism to promote planar divisions, or (2) planar may be the default process.

We favor the latter hypothesis for the following reasons. First, LGN is necessary for NuMA's cortical localization, and NuMA—or dynactin— loss leads to a similar phenotype as LGN loss in the epidermis (*Seldin et al., 2016*; *Seldin et al., 2013*; *Williams et al., 2011*). While there are numerous examples in the literature where NuMA can direct spindle orientation in an LGN/Pins-independent manner (*Bergstralh et al., 2016*; *Bosveld et al., 2016*; *Kiyomitsu and Cheeseman, 2013*; *Kotak et al., 2014*), there are few examples where spindles can be actively reoriented independently of NuMA and dynein. Second, planar divisions predominate in the early epidermis, prior to and during early stratification, and we are not familiar with any genetic alteration that induces *precocious* perpendicular divisions.

More likely, we believe that planar divisions are a default state, attributable to high tension across the basal layer, possibly as a result of stronger adhesive forces on lateral vs. apical cell membranes, or intra-tissue tension provided by differentiated layers (*Ning et al., 2021*). Of interest, Devenport et al. recently described increased perpendicular divisions and hyperthickened epidermis in *Vangl2* mutants (*Box et al., 2019*). Unexpectedly, this phenotype was not due to defective planar cell polarity, but rather to tissue-wide changes in cell shape and packing caused by their neural tube defect. In addition to this effect of interphase cell shape on division orientation, hypoproliferation, and elevated apoptosis can lead to non-cell autonomous increases in planar divisions (*Morrow et al., 2019*; *Soffer et al., 2022*). Future studies will be necessary to directly test how local changes in cell density, and intra-tissue tension, impact oriented cell divisions, and the balance between proliferation and differentiation.

## Coordinating self-renewal and differentiation on a tissue scale

In the epidermis, ACDs are one of two known mechanisms to promote differentiation, the other being delamination (*Watt and Green, 1982*). Until 15 years ago, delamination—for example, differentiation

by detachment rather than division—was thought to be the driving force for epidermal differentiation (*Blanpain and Fuchs, 2006*). While this is true during adult epidermal homeostasis, where perpendicular divisions are rare to non-existent (*Clayton et al., 2007*; *Ipponjima et al., 2016*; *Mesa et al., 2018*; *Rompolas et al., 2016*), it is now clear that ACDs are essential for proper skin development in the embryo (*Lechler and Fuchs, 2005*; *Seldin et al., 2016*; *Williams et al., 2014*). In the adult, the processes of proliferation and differentiation cooperate to regulate cell density in the basal layer, such that delamination precedes and induces self-renewal in nearby cells (*Cockburn et al., 2022*; *Mesa et al., 2018*). Interestingly, in the embryo, the opposite relationship exists, such that proliferation seems to drive neighboring cells to delaminate (*Miroshnikova et al., 2018*). In other tissues, local crowding drives live-cell extrusion (*Eisenhoffer et al., 2012*; *Marinari et al., 2012*), and 'winners' (stem cells) and 'losers' (differentiating cells) have also been described during epidermal stratification (*Ellis et al., 2019*). Yet, whether the two differentiation processes of ACD and delamination are linked has not been explored. Here, using genetic lineage tracing, we find that delamination increases when ACDs decrease in LGN KOs. It will be of interest to see how these global changes are influenced by the local tissue microenvironment.

## Materials and methods

### Animals
All mice were housed in an AAALAC-accredited (#329; November 2020), USDA registered (55 R-0004), and NIH welfare-assured (D16-00256 (A3410-01)) animal facility. All procedures were performed under IACUC-approved animal protocols (19-155 and 22-121). For fixed sample imaging (immunohistochemistry) and all lentiviral transduction experiments (unless otherwise noted), CD1 WT outbred mice (Charles Rivers; #022) were utilized. *Gpsm2*$^{-/-}$ KO (*Gpsm2tm1a(EUCOMM)Wtsi*; Jackson Labs #4441912 via Basile Tarchini) (*Tarchini et al., 2013*) mice were maintained on a mixed C57BL6/J background and bred to either (1) *Krt14*$^{CreER}$; *Rosa26*$^{Confetti}$ (*Tg(KRT14-cre/ERT)20Efu*; Jackson Labs #005107/*Gt(ROSA)26Sortm1(CAG-Brainbow2.1)Cle*; Jackson Labs #013731) females or identical males for genetic lineage tracing, and (2) *Rosa26*$^{mTmG}$ (*Gt(ROSA)26Sortm4(ACTB-tdTomato,-EGFP)Luo/J*; Jackson Labs #007576) homozygous females with at least one copy of the *Krt14*$^{Cre}$ allele (crossed to males of the identical genotype), for ex vivo live imaging. *Gpsm1*$^{-/-}$ KO mice (*AGS3DEL(B6.129S6(SJL))-Gpsm1tm1.1Lajb/J*; Jackson Labs # 019503 via Ricardo Richardson) (*Blumer et al., 2008*) were maintained on a mixed 129S6 background and bred to the same strains as *Gpsm2* KOs for lineage tracing and live-imaging. *Gpsm1*$^{-/-}$; *Gpsm2*$^{-/-}$ mice were maintained on a mixed background and bred to the same strains as *Gpsm2* KOs for lineage tracing and live imaging. CD1, *Gpsm2* KO, and *Rosa26*$^{mT/mG}$; *Krt14*$^{Cre}$ animals were injected with lentiviral constructs (see below). Note that all ages are defined where E0.5 is noon of the day that a plug is found, but that developmental differences exist between strains. For example, we have found the 129S6 background of AGS3 KOs to be ~1 day behind, and the C57BL6/J background of LGN KOs to be ~0.5 day behind outbred CD1s.

### Live imaging
The protocol for live imaging has been adapted from the technique described by the Devenport lab (*Cetera et al., 2018*). For a full protocol please see *Lough et al., 2019*. Briefly, epidermal explants were harvested from the mid-back of WT E16.5 *Rosa26*$^{mTmG}$; *Krt14*$^{Cre}$ embryos, crossed to either *Gpsm2* KO, *Gpms1* KO, or double mutant background. The explants were sandwiched between a gel/media patty and a gas-permeable membrane dish, and were cultured at 37°C with 5.0% $CO_2$ for >1.5 hr prior to- and throughout the course of imaging. Confocal imaging was used to acquire a 20–30 μm z-stack every 5 min for 3–6 hr in a temperature controlled chamber, with the exception of images in *Figure 2E, F*, which were acquired using the Dragonfly spinning disk confocal (Andor) equipped with a Leica ×40/1.4 NA Oil Plan Neo objective. Images were acquired with 5-min intervals and a Z-series with 0.5-mm step size (total depth ranging from 46 μm) for 6 hr. The Andor iXon life 888 BV was used to image mKate2, Andor Zyla 4.2 Plus was used for imaging mGFP. Additionally, we performed live imaging of E16.5 lentiviral-transduced *Gpsm1*$^{1147}$ H2B-mRFP1 epidermal explants on a *Rosa26*$^{mTmG}$; *Krt14*$^{Cre}$ background. Divisions appearing close to the tissue edge or showing any signs of disorganization/damage were avoided to exclude morphological changes associated with wound repair. 4D image sets were processed with a DL image denoising method using self-supervised training called

N2V (detailed bellow), and processed using ImageJ (Fiji). Measurements of division orientation at $t = 0$ and $t = 60$ determined from movie stills are provided in the accompanying source data file.

## Lentiviral injections

The protocol for lentiviral injection was performed according to *Beronja et al., 2010*, under approved IACUC protocol 19-155. Pregnant mice mice were anesthetized for less than 1 hr and provided subcutaneous analgesics (5 mg/kg meloxicam and 1–4 mg/kg bupivacaine). A uterine horn was pulled out the mom into a phosphate-buffered saline (PBS) filled culture dish to expose E9.5 embryos. We performed a microinjection of ~0.7 μl of concentrated lentivirus into the amniotic space using a custom glass needle that was visualized by ultrasound. Three to six embryos were injected on the same horn per pregnant dam, and the non-injected horn was used for matched littermate controls. Following injection, the horns were put back into the thoracic cavity of the dam and sutured closed. Surgical staples were used to reseal the skin incision. Once awake and freely moving, the dam was monitored for 4–7 days. Embryos were harvested and processed at E14.5–E17.5.

## Genetic lineage tracing

For full protocol, please see *Lough et al., 2019*. Males of identical genotype were crossed with $Krt14^{CreERT2}$; $Rosa26^{Confetti}$ females with either *Gpsm2* KO, *Gpsm1* KO or double mutant genotypes. A 100 μg per gram dam mass of tamoxifen was delivered by oral gavage at E15.5 to activate the $Krt14^{CreERT2}$ allele. Following tamoxifen dosing dams were monitored for 24 hr for signs of abortion or distress. Forty-eight hours after tamoxifen delivery, embryos were harvested at E17.5, backskins were fixed for 30 min in 4% paraformaldehyde and washed with PBS. Fixed backskins were embedded in OCT and sectioned sagittally at 8 μm. To enhance the fluorophores of the $Rosa26^{Confetti}$ allele we immunostained backskin sections (see below), omitting the 5-min post-fixation step, using monoclonal Rat anti-mCherry clone 16D7 (Life Technologies M11217) to enhance membrane RFP, and polyclonal Chicken anti-GFP (Abcam ab13970) to enhance the membrane-CFP, nuclear-GFP, and cytoplasmic-YFP fluorophores. Areas of the stained section with labeled clones were acquired with a ×40/1.15 NA objective with ×1.5 digital zoom. We scored sparse clones (<1% total cells) for the number of suprabasal cells (distinguished by staining with anti-Krt10 antibody), and basal cells (distinguished by staining with anti-Krt14 antibody). To exclude the possibility that tamoxifen induction occurred while cells were already suprabasally positioned, we only counted delamination events as clones with suprabasal cells in the stratum spinosum (SS) layer.

## Constructs and RNAi

For *Gpsm2* and *Gpsm1* RNAi targeting, we utilized shRNAs that had been previously validated (*Williams et al., 2011*; *Williams et al., 2014*). The nucleotide base (NCBI Accession number) for a given shRNA clone are identified by the gene name followed by the 21-nucleotide target sequence (e.g., $Gpsm1^{1147}$). Packaging of lentivirus was performed using 293 FT cells and pMD2.G and psPAX2 helper plasmids (Addgene plasmids #12259 and #12260, respectively). To evaluate shRNAs for their knockdown efficiency, we infected primary keratinocytes with a MOI of ~1 in E-Low calcium medium for ~48 hr. Puromycin selected cell lines were lysed with RNeasy Mini Kit (QIAGEN) to isolate RNA. mRNA knockdown efficiency was determined by RT-qPCR using two independent primer sets for each transcript with *Hprt1* and cyclophilin B (*Ppib2*) as reference genes and cDNA from stable cell lines expressing Scramble shRNA as a reference control. The following primer sequences were used: Scramble (5'-CAACAAGATGAAGAGCACCAA-3') and $Gpsm1^{1147}$ (5'- GCCTTGACCTTTGCCAAGAA A-3'). Sequences for *Hprt1* and *Ppib2* primers have been described previously (*Williams et al., 2011*). All materials are available from the corresponding author upon request.

## Antibodies, immunohistochemistry, and fixed imaging

E15.5–E16.5 embryos were skinned and flat-mounted on Whatman paper. E14.5 embryos were mounted whole. In all cases (except for lineage tracing experiments), samples were mounted in OCT (Tissue Tek) and frozen fresh at −20°C. Experimental genotypes (homozygous, heterozygous, and WT) were mounted in the same block to allow for direct comparisons on the same slide. Similarly, infected and uninfected littermate controls were mounted in the same blocks. Frozen samples were sectioned (8 μm thickness) on a Leica CM1950 cryostat. Staining was conducted as previously

described (*Lough et al., 2019*). Images were acquired using LAS AF software on a Leica TCS SPE-II 4 laser confocal system on a DM5500 microscope with a ACS Apochromat ×40/1.15 NA oil, or ACS Apochromat ×63/1.30 NA oil objectives. The following primary antibodies were used: Survivin (rabbit mAb 71G4B7, Cell Signaling 2808S, AB_2063948, 1:500), LGN (guinea pig, 1:500) (*Williams et al., 2011*), LGN (rabbit, Millipore ABT174, AB_2916327, 1:2000) (*Williams et al., 2014*), phospho-histone H3 (rat, Abcam AB10543, AB_2295065, 1:1000–5000), β4-integrin (rat, ThermoFisher 553745, AB_395027, 1:1,000), Cytokeratin-14 (chicken, Biolegend 906004, AB_2616962, 1:5000), Cytokera-tin-14 (guinea pig, AB_979615, Origene BP5009, 1:1000), Cytokeratin-10 (rabbit, BioLegend 905401, AB_2565049, 1:1000), GFP (chicken, Abcam AB13970, AB_300798, 1:1000), V5 (chicken, Abcam AB9113, AB_307022, 1:1000), mCherry (rat, Life Technologies M11217, AB_2536611, 1:1000–3000), RFP (rabbit, MBL PM005, AB_591279, 1:1000), RFP (chicken, Millipore AB3528, AB_91496, 1:500), and TagRFP (rabbit polyclonal, Thermo Fisher R10367, AB_10563941, 1:1000). The following secondary antibodies were used (all antibodies produced in donkey): anti-rabbit AlexaFluor 488 (Life Technologies, 1:1000), anti-rabbit Rhodamine Red-X (Jackson Labs, 1:500), anti- rabbit Cy5 (Jackson Labs, 1:400), anti-rat AlexaFluor 488 (Life Technologies, 1:1000), anti-rat Rhodamine Red-X (Jackson Labs, 1:500), anti-rat Cy5 (Jackson Labs, 1:400), anti-guinea pig AlexaFluor 488 (Life Technologies, 1:1000), anti-guinea pig Rhodamine Red-X (Jackson Labs, 1:500), anti-guinea pig Cy5 (Jackson Labs, 1:400), anti-goat AlexaFluor 488 (Life Technologies, 1:1000), anti-goat Cy5 (Jackson Labs, 1:400), anti-mouse IgG AlexaFluor 488 (Life Technologies, 1:1000), anti-mouse IgG Cy5 (Jackson Labs, 1:400), anti-mouse IgM Cy3 (Jackson Labs, 1:500). For some multi-labeling experiments where Survivin immunostaining was used in combination with another rabbit primary antibody, Alexa488 conjugated rabbit mAb anti-Survivin (71G4B7) (Cell Signaling 2810, AB_10691462, 1:1000) was used and added after all other primary and secondary antibodies were added to avoid cross-reactivity.

## Image processing
### Denoising
Live-imaging acquisition on epidermal explants was conducted on a Zeiss LSM 710 Spectral confocal laser scanning microscope as detailed in *Lough et al., 2019*. To restore noisy images we employed N2V, a DL image denoising method using self-supervised training (*Krull et al., 2020*). While this method does not require ground truth or low noise equivalent images to train a DL model, we never-theless validated our N2V models by error mapping and quality metrics estimation of ground truth images (SSIM and RSE maps).

## Spindle and division orientation
Spindle orientation measurements was performed as in *Byrd et al., 2016*; *Lough et al., 2019*; *Williams et al., 2011*; *Williams et al., 2014*, using Survivin as a marker of late-stage mitotic cells, with angles between daughter nuclei relative to the basement membrane measured using ImageJ. Rare anaphase cells were characterized by broadly distributed Survivin staining between daughter cells, while telophase cells showed dual-punctate Survivin staining. Division orientation measurements were calculated as the angle between a vector parallel to the basement membrane and a vector connecting the estimated center of each daughter nuclei. Cells where two daughter nuclei surrounding punctate Survivin staining could not be unambiguously identified were not counted. In cases where the division angle was oblique in the z-plane, image stacks were acquired to facilitate measurement of the division angle. Manual annotation tools in the LAS X software were used to draw arrows indicating daughter cells at the time of imaging to ensure that the correct angle was measured later in ImageJ.

Similarly, division orientation was measured in live-imaging experiments, where the position of the daughter nuclei was inferred based on cell volume/shape changes and/or the presence of the H2B-mRFP1 nuclear reporter (in AGS3 KD experiments). All divisions orientation measurements are provided in the source data file.

## LGN/AGS3 localization/intensity
Imaging of CD1 lentiviral H2B-RFP animals was performed with control littermates and experimental samples on the same slide to avoid variation in antibody staining. Similarly, with experimental geno-types, imaging was performed with control littermates on the same slide. Scoring of LGN localization patterns (e.g., apical, unpolarized, absent) was determined for cells labeled with pHH3 or Survivin

(telophase marker), irrespective of the lentiviral H2B-RFP reporter to avoid bias. For linescan measurements of LGN cortical intensity (see *Figure 4C–E*), similar methodology was utilized as described in *Lough et al., 2019*, using E-cadherin to define the boundary of the cell cortex in mitotic cells from E16.5 mosaically transduced *Gpsm1[1147]* H2B-mRFP1 epidermis. To support comparisons across cells with variable circumferences, linescan data were compressed into 100 data points relative to the proportion of total line length. All mitotic cells in which a detectable LGN signal could be observed were quantified, and were later binned into AGS3 KD (RFP+) or WT (RFP−) based on the presence of absence of the nuclear H2B-mRFP1 reporter, which labels transduced cells. Measurements of non-cortical (cytoplasmic) fluorescence intensity were used for background subtraction. Mean (± standard deviation) cytoplasmic background values were equivalent in RFP− (11.6 ± 4.9 AU) and RFP+ (12.4 ± 4.6 AU) populations. For determination of the threshold to be used for calculating crescent width and integrated F.I., 10, 20, and 30 AU were each empirically tested, and yielded p-values of 0.302, 0.023, and 0.041, respectively, by Mann–Whitney test in comparing RFP− and RFP+ populations. Data from the 20 AU threshold were chosen for display in *Figure 4* because this was the minimum value that eliminated most isolated (minor) peaks in fluorescence, which we considered to be noise. Cortical Coverage was determined as the percentage of the contiguous portion of the LGN F.I. plot that exceeded the 20 AU threshold. Integrated F.I. (referred to as AUC) was calculated as the aggregate area of trapezoids $((a + b)/2)$ for each unit of the area defined by Cortical Coverage, where $a$ and $b$ are defined as threshold-subtracted individual measurements of LGN F.I.

## Stratification

Epidermal thickness was measure in E14.5–E17.5 embryos by staining for H2B-RFP, K14, and K10. We imaged >10 regions of sagittal backskin sections per quantified embryo. We thresholded the K10 signal and created a binary mask that was used to quantify the suprabasal area above threshold. The area was then divided by the length of the underlying basement membrane (determined by K14 staining). *n* values for these analyses are representative of the number of regions imaged.

## Statistical analyses and graphs

Error bars represent standard error of the mean unless otherwise noted. Statistical tests of significance were determined by Mann–Whitney *U*-test (non-parametric) or Student's *t*-test (parametric) depending on whether the data fit a standard distribution (determined by pass/fail of majority of the following: Anderson–Darling, D'Agostino and Pearson, Shapiro–Wilk, and Kolmogorov–Smirnov tests). Cumulative frequency distributions were evaluated for significant differences by Kolmogorov–Smirnov test. All analyses were generated using GraphPad Prism 9. Figures were constructed using Fiji and Adobe Illustrator CC 2021. Sample sizes were chosen so as to meet or exceed those used in previous studies where power analyses were conducted using an $\alpha$-value of 0.05 and power of 80% (*Lough et al., 2019*). All data measurements related to the graphs are provided in the source data spreadsheet.

## Acknowledgements

We thank Basile Tarchini (Jackson Laboratories) for sharing the *Gpsm2* mice and Ricardo Richardson (North Carolina Central University) and Steve Lanier (Wayne State University) for sharing the *Gpsm1* mice. We are grateful to Ali Vural (Wayne State University) for sharing AGS3 antibodies and for helpful discussions. We thank Juliet King, Bethany Brown, and Kevin Byrd for helpful discussions and technical support, as well as Bob Goldstein, Joan Taylor, Jim Bear, and Stephanie Gupton for critical reading of the manuscript. We thank Jabiz Nafisi and Carien Niessen for their insights and sharing unpublished data. We are grateful to Pablo Ariel and the Microscopy Services Laboratory (MSL) core facility for assistance and guidance with live imaging and image processing. These studies were supported by grants from the NIH R01 AR077591 (SEW), Binational Science Foundation 2019230 (SEW), Sidney Kimmel Foundation Kimmel Scholars Program SKF-15-065 (SEW), and Chan Zuckerberg Initiative DAF, an advised fund of Silicon Valley Community Foundation grant 2020-225716 (KMK). The MSL is supported in part by P30 CA016086 Cancer Center Core Support Grant to the UNC Lineberger Comprehensive Cancer Center, and the Andor Dragonfly microscope was funded with support from NIH grant S10 OD030223.

# Additional information

## Funding

| Funder | Grant reference number | Author |
|---|---|---|
| National Institutes of Health | R01 AR077591 | Scott E Williams |
| United States - Israel Binational Science Foundation | 2019230 | Scott E Williams |
| Sidney Kimmel Foundation | SKF-15-065 | Scott E Williams |
| Chan Zuckerberg Initiative | 2020- 225716 | Katarzyna M Kedziora |

The funders had no role in study design, data collection, and interpretation, or the decision to submit the work for publication.

## Author contributions

Carlos P Descovich, Conceptualization, Data curation, Formal analysis, Validation, Investigation, Visualization, Methodology, Writing - original draft, Writing - review and editing; Kendall J Lough, Conceptualization, Formal analysis, Methodology, Writing - review and editing; Akankshya Jena, Jessica J Wu, Jina Yom, Manuela Uppalapati, Investigation; Danielle C Spitzer, Formal analysis, Investigation; Katarzyna M Kedziora, Software, Formal analysis, Methodology, Writing - review and editing; Scott E Williams, Conceptualization, Formal analysis, Supervision, Funding acquisition, Validation, Investigation, Visualization, Methodology, Writing - original draft, Project administration, Writing - review and editing

## Author ORCIDs

Carlos P Descovich (iD) http://orcid.org/0000-0002-6366-5195
Danielle C Spitzer (iD) http://orcid.org/0000-0003-4827-1857
Scott E Williams (iD) http://orcid.org/0000-0001-9975-7334

## Ethics

This study was performed in strict accordance with the recommendations in the Guide for the Care and Use of Laboratory Animals of the National Institutes of Health. All of the animals were handled according to approved Institutional Animal Care and Use Committee (IACUC) protocols 19-155 and 22-121 at the University of North Carolina. All mice were housed in an AAALAC-accredited (#329; November 2020), USDA registered (55-R-0004), and NIH welfare-assured D16-00256 (A3410-01) animal facility. All surgeries were performed under isoflurane anesthesia and meloxicam was alleviated post-operatively to minimize pain.

## Decision letter and Author response

Decision letter https://doi.org/10.7554/eLife.80403.sa1
Author response https://doi.org/10.7554/eLife.80403.sa2

# Additional files

## Supplementary files

• MDAR checklist

• Source data 1. Source data contains raw data measurements for all figure panels with data quantification.

## Data availability

All data generated or analyzed during this study are included in the manuscript and supporting file; Source Data files have been provided for data in all figures.

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
