## [Editor Report]

Descovich et al. examine the important decision between proliferative (planar) and differentiation (perpendicular) divisions in the basal layers of the skin and find a key promoter of perpendicular divisions is inhibited by its paralog to specify planar divisions. The authors use sophisticated mouse genetics and imaging and discover that LGN and its paralog AGS3 function antagonistically in determining perpendicular vs. planar divisions. The claim that AGS3 functions as a natural dominant negative version of LGN to control division orientation is well supported.

---

## [Decision Letter]

**Decision letter after peer review:**

Thank you for submitting your article "AGS3 antagonizes LGN to balance oriented cell divisions and cell fate choices in mammalian epidermis" for consideration by *eLife*. Your article has been reviewed by 3 peer reviewers, including Yukiko M Yamashita as the Reviewing Editor and Reviewer #1, and the evaluation has been overseen by Anna Akhmanova as the Senior Editor. The following individuals involved in the review of your submission have agreed to reveal their identity: Dan Bergstralh (Reviewer #2); Kenneth E Prehoda (Reviewer #3).

Essential revisions:

As you can see in the individual reviewers' comments, they are positive overall. However, they commonly identified some critical issues. Among those, the most important issue is about the localization of LGN: its arbitrary calling (polarized, not-polarized, absent) is not convincing. A better quantification must be used to determine the localization of LGN upon manipulation of AGS3. Details are explained in individual comments. Reviewers also made other important suggestions.

*Reviewer #1 (Recommendations for the authors):*

Throughout the manuscript, it is a bit hard to keep straight between LGN, AGS3, Gpsm1, and Gpsm2. It might be easier for readers to follow if you always put the names together, as in Figure 1 title ('LGN/Gpsm2').

*Reviewer #2 (Recommendations for the authors):*

Recommendations/Concerns:

Division angles in Gpsm2-/- tissue are obviously (and statistically) different from those observed in wild-type tissue. However, since 16% of the divisions (about 20) are perpendicular, the claim that LGN is "necessary" for perpendicular divisions (line 134) seems incorrect. I suggest this word could be replaced with "important." The finding that perpendicular divisions are decreased but not eliminated should be discussed in the text since it suggests that either the measurements are imperfect (this seems plausible) or that the LGN-dependent mechanism promotes – but is not required for – perpendicular division.

Experiments presented in Figure 2 are exceptionally thorough and careful.

Figure 3B is critical to the paper. At present, it is under-described and not entirely convincing.

I struggled to understand what the color scheme in Figure 3B is meant to convey. There are three colors associated with "apical," but the distinction between these three is not explained in the legend. Once I did understand it, I couldn't see why it was useful. This issue persists in the remaining figures.

How is "absent" distinguished from "unpolarized" in 3B? I didn't find a description in the text or figure, though my interpretation of the relevant Methods section is that the determination was made manually. Could "absent" simply reflect low signal intensity or detection? Presumably, it is easiest to observe LGN when it is concentrated (apically), whereas an unpolarized (broadly cortical) signal is more difficult to detect. If the signal is low, might unpolarized and absent be the same thing? If not, then what does absent vs. unpolarized mean? Is AGS3 preventing cortical localization of LGN, polarized enrichment of LGN, or both? One important reason to clarify this is that AGS3 overexpression appears to markedly decrease the number of cells in the "absent" category. Gpsm1 knockdown doesn't have the opposite effect. Another important reason is that significance is determined using chi-square, meaning presumably that the proportion of cells in the absent and unpolarized categories (as opposed to just "apical" vs "not apical") is taken into account when determining significance.

If possible, quantification of pixel intensity (A) around the cell perimeter and (B) in comparison to the cytoplasm could be a helpful strategy for making the distinction.

The discussion is scholarly and thorough. It was a pleasure to read.

The authors suggest that AGS3 could work by displacing LGN from a binding partner. Firstly, the finding that overexpressed AGS3 impacts LGN only after metaphase suggests that such an interaction must be temporally regulated, perhaps through post-translational modification. (A trivial suggestion is that the authors could raise this and/or other possibilities in the discussion.) Secondly, and more importantly, the authors suggest that the relevant binding partner is NuMA (line 501). In making this claim it seems that the authors are ascribing a new function to NuMA, namely the regulation of Telophase Correction. I am unaware of any data linking NuMA to this process and I am a bit confused as to why it would be, since cortical localization of NuMA at anaphase is reported to be LGN-independent.

As described in the Discussion, Culurgioni and Mapelli (2013) indeed suggested that Insc recruits LGN to the apical cortex, then hands it off to NuMA. In a subsequent paper not cited in this manuscript, Culurgioni et al. (2018) propose instead that LGN and Insc exist in a separate stable pool to LGN and NuMA, and that hand-off does not occur.

*Reviewer #3 (Recommendations for the authors):*

I found the abstract and introduction a bit confusing because of statements like, "Little is known about the molecular regulation of planar divisions". My understanding is the planar divisions are known to be molecularly regulated by the lack of cortical LGN, so the real question is what makes only some cells have cortical LGN. This confused me because at times I thought the paper was going to be about the specific mechanisms that make planar divisions planar, which the discussion speculates on, but are not the main topic of the paper.

I had difficulty seeing the reorientation in Figure 4 supplement 1 (for that matter, the ones in Figures 4B, and C are difficult to see, even with the angles drawn on them). Perhaps fewer frames or some markings in the supplement could be beneficial to readers.

Figure 1D suggests statistical test is between AGS3 OE and KD -must be between AGS3 OE and WT.

---

## [Author Response]

Reviewer #1 (Recommendations for the authors):Throughout the manuscript, it is a bit hard to keep straight between LGN, AGS3, Gpsm1, and Gpsm2. It might be easier for readers to follow if you always put the names together, as in Figure 1 title ('LGN/Gpsm2').

Throughout the text and figures we have made changes to make it easier to understand when we are referring to AGS3/Gpsm1 and LGN/Gpsm2. We now use terms such as AGS3 knockout (KO) in place of *Gpsm1^-/-^*, AGS3 knockdown (KD) in place of *Gpsm1^1147^*, and similar terms for LGN in lieu of *Gpsm2*.

Reviewer #2 (Recommendations for the authors):Recommendations/Concerns:Division angles in Gpsm2-/- tissue are obviously (and statistically) different from those observed in wild-type tissue. However, since 16% of the divisions (about 20) are perpendicular, the claim that LGN is "necessary" for perpendicular divisions (line 134) seems incorrect. I suggest this word could be replaced with "important." The finding that perpendicular divisions are decreased but not eliminated should be discussed in the text since it suggests that either the measurements are imperfect (this seems plausible) or that the LGN-dependent mechanism promotes – but is not required for – perpendicular division.

We agree and have changed the word “required” to “important” in line 130. We also include a sentence in the Discussion to speculate that residual perpendicular divisions observed in LGN KOs could be attributable to an LGN-independent mechanism of NuMA cortical recruitment (lines 536-539).

Experiments presented in Figure 2 are exceptionally thorough and careful.

Thanks! We feel this was an important point and since we lacked the ideal tool (an AGS3-specific antibody to detect endogenous protein) we made efforts to demonstrate AGS3 localization through a variety of orthogonal approaches.

Figure 3B is critical to the paper. At present, it is under-described and not entirely convincing.

We agree, and as described above, have undertaken several new studies to quantify LGN fluorescent intensity in Figure 4C-J.

I struggled to understand what the color scheme in Figure 3B is meant to convey. There are three colors associated with "apical," but the distinction between these three is not explained in the legend. Once I did understand it, I couldn't see why it was useful. This issue persists in the remaining figures.

This color scheme was our attempt to make it easier to follow different genotypes throughout the paper (e.g. AGS3 KO was always shown in blue, LGN KO in red, AGS3 + LGN KO in magenta). However we understand how the multiple labeling of bar graphs could be confusing, so we stuck with the simple black/white/gray color scheme for this type of graph. Instead, different genotypes were color-coded in the label text when multiple genotypes are displayed in the same bar graphs (e.g. Figure 7G-I). We hope this is more clear.

How is "absent" distinguished from "unpolarized" in 3B? I didn't find a description in the text or figure, though my interpretation of the relevant Methods section is that the determination was made manually. Could "absent" simply reflect low signal intensity or detection? Presumably, it is easiest to observe LGN when it is concentrated (apically), whereas an unpolarized (broadly cortical) signal is more difficult to detect. If the signal is low, might unpolarized and absent be the same thing? If not, then what does absent vs. unpolarized mean? Is AGS3 preventing cortical localization of LGN, polarized enrichment of LGN, or both? One important reason to clarify this is that AGS3 overexpression appears to markedly decrease the number of cells in the "absent" category. Gpsm1 knockdown doesn't have the opposite effect. Another important reason is that significance is determined using chi-square, meaning presumably that the proportion of cells in the absent and unpolarized categories (as opposed to just "apical" vs "not apical") is taken into account when determining significance.If possible, quantification of pixel intensity (A) around the cell perimeter and (B) in comparison to the cytoplasm could be a helpful strategy for making the distinction.

This is a valid point, since determination of LGN localization pattern in these experiments was a qualitative assessment. We have added new analyses, as suggested by the Reviewer, where we quantify LGN pixel intensity at the perimeter of the cell cortex compared to the cytoplasm.

The discussion is scholarly and thorough. It was a pleasure to read.As described in the Discussion, Culurgioni and Mapelli (2013) indeed suggested that Insc recruits LGN to the apical cortex, then hands it off to NuMA. In a subsequent paper not cited in this manuscript, Culurgioni et al. (2018) propose instead that LGN and Insc exist in a separate stable pool to LGN and NuMA, and that hand-off does not occur.

We appreciate this compliment as well as the suggestion to amend our interpretation of the Mapelli model. We have further elaborated on this point in the Discussion (lines 489-494).

The authors suggest that AGS3 could work by displacing LGN from a binding partner. Firstly, the finding that overexpressed AGS3 impacts LGN only after metaphase suggests that such an interaction must be temporally regulated, perhaps through post-translational modification. (A trivial suggestion is that the authors could raise this and/or other possibilities in the discussion.) Secondly, and more importantly, the authors suggest that the relevant binding partner is NuMA (line 501). In making this claim it seems that the authors are ascribing a new function to NuMA, namely the regulation of Telophase Correction. I am unaware of any data linking NuMA to this process and I am a bit confused as to why it would be, since cortical localization of NuMA at anaphase is reported to be LGN-independent.

These are insightful comments, and we have amended the Discussion to address them. Since we are now able to quantitatively show that AGS3 loss enhances LGN’s cortical recruitment, we no longer believe that it functions exclusively during telophase correction. However, we discuss how post-translational mechanisms may regulate the activity of LGN/AGS3 and their affinity for binding partners (e.g. lines 492-494, 597-619). As mentioned earlier, we have added new text to the Discussion where we explicitly mention data in other systems that support the existence of LGN-independent mechanisms of NuMA cortical recruitment (lines 533-544).

Reviewer #3 (Recommendations for the authors):I found the abstract and introduction a bit confusing because of statements like, "Little is known about the molecular regulation of planar divisions". My understanding is the planar divisions are known to be molecularly regulated by the lack of cortical LGN, so the real question is what makes only some cells have cortical LGN. This confused me because at times I thought the paper was going to be about the specific mechanisms that make planar divisions planar, which the discussion speculates on, but are not the main topic of the paper.

We agree with this point and have altered the wording in the abstract (line 8) and introduction (lines 36-37, 47-49) to better emphasize that our studies address how LGN becomes polarized in some cells and not others, and the role AGS3 plays in this process.

I had difficulty seeing the reorientation in Figure 4 supplement 1 (for that matter, the ones in Figures 4B, and C are difficult to see, even with the angles drawn on them). Perhaps fewer frames or some markings in the supplement could be beneficial to readers.

We have added annotations to the supplemental figure panels (now Figure 5), which we hope will be helpful.

Figure 1D suggests statistical test is between AGS3 OE and KD -must be between AGS3 OE and WT.

Thank you for catching this error. The statistical test was indeed conducted between these two groups and this was mistake in the placement of the bar. However, since we have removed the overexpression data from this graph and placed it in Figure 3B this is no longer relevant.